# IFNγ signaling in cytotoxic T cells restricts anti-tumor responses by inhibiting the maintenance and diversity of intra-tumoral stem-like T cells

Julie M. Mazet[1], Jagdish N. Mahale[1], Orion Tong [2,3], Robert A. Watson [2,3], Ana Victoria Lechuga-Vieco [1], Gabriela Pirgova[1], Vivian W. C. Lau[1], Moustafa Attar [1], Lada A. Koneva[1], Stephen N. Sansom [1], Benjamin P. Fairfax [2] & Audrey Gérard [1] ✉

IFNγ is an immune mediator with concomitant pro- and anti-tumor functions. Here, we provide evidence that IFNγ directly acts on intra-tumoral CD8 T cells to restrict anti-tumor responses. We report that expression of the IFNγ receptor β chain (IFNγR2) in CD8 T cells negatively correlates with clinical responsiveness to checkpoint blockade in metastatic melanoma patients, suggesting that the loss of sensitivity to IFNγ contributes to successful anti-tumor immunity. Indeed, specific deletion of IFNγR in CD8 T cells promotes tumor control in a mouse model of melanoma. Chronic IFNγ inhibits the maintenance, clonal diversity and proliferation of stem-like T cells. This leads to decreased generation of T cells with intermediate expression of exhaustion markers, previously associated with beneficial anti-tumor responses. This study provides evidence of a negative feedback loop whereby IFNγ depletes stem-like T cells to restrict anti-tumor immunity. Targeting this pathway might represent an alternative strategy to enhance T cell-based therapies.

Tumors actively escape the immune system[1] by inducing an immu-nosuppressive state[2], in which CD8 tumor-infiltrating lymphocytes (TILs) are characterized by diminished effector functions[3]. Pro-grammed cell death-1 (PD-1) and cytotoxic T-lymphocyte-associated protein 4 (CTLA-4) are two checkpoint receptors often expressed on TILs that are frequently co-opted by tumors to dampen effector functions when engaged by their ligands, PDL-1 and B7[4]. Although the blockade of these checkpoint molecules has shown unprece-dented success in treating aggressive cancers[5,6], its response rates are approximately 15 to 30%[7]. The limited efficacy of checkpoint blockade suggests that additional factors restrict immunity to tumors, illustrating the complex crosstalk between tumor and immune cells[8].

To understand the different factors that might be at play, numerous groups have studied the T cell subsets found in blood and tumors from cancer patients and their correlation with response to checkpoint blockade. Durable clinical benefit following checkpoint blockade in patients with metastatic melanoma is associated with the presence of large T cell clones, defined as CD8 T cells expressing the same T cell receptor (TCR) and occupying more than 0.5% of the T cell repertoire in blood[9,10]. In addition, a specific T cell subset with stem-like properties, characterized by Tcf7, Ccr7, and IL7R expression and intermediate expression of PD-1, is key to the response to checkpoint blockade in chronic infection and cancer[11–14], and for enhanced immune cytotoxic response in chronic conditions in both mice and humans[12–14]. Stem-like T cells mediate the proliferative response to

[1]The Kennedy Institute of Rheumatology, University of Oxford, Oxford, UK. [2]Department of Oncology, University of Oxford, Oxford, UK. [3]These authors contributed equally: Orion Tong, Robert A. Watson. ✉e-mail: Audrey.gerard@kennedy.ox.ac.uk

checkpoint blockade, generating both stem-like and differentiated/ exhausted cells. Ablation of stem-like T cells restricts responses to immunotherapy[14]. While the relevance of stem-like T cells for tumor immunity is well described, their regulation is not completely understood. There is an emerging concept that stem-like T cells are maintained in lymph nodes (LNs), replenishing the tumor pool that quickly becomes depleted due to chronic antigen stimulation[15,16]. Although stem-like T cells are also found within tumor sites[14], they lack comparable self-renewal potential and maintenance compared to LN stem-like T cells, the reason for which remains unclear.

Tumor immunity relies on immune mediators, such as the cytokine IFNγ[17,18]. IFNγ-related gene signature is a predictive marker for chemotherapy, radiotherapy and immunotherapy efficacy in multiple tumor types[18]. Mechanistically, IFNγ inhibits tumor proliferation[19], and angiogenesis[20]. However, several lines of evidence also point to IFNγ as a factor restricting antitumor immunity[21,22] by promoting tumor initiation, growth and immune evasion[23,24]. As a result, cancer therapies based on recombinant IFNγ treatment are rarely effective[25]. Because IFNγ is inherently linked to the efficacy of tumor immunity, it is critical to fully understand how it can also simultaneously impede tumor immunity[26]. While most studies have focused on the impact of IFNγ on cancer and stromal cells, the consequences of IFNγ on immune cells, in particular T cells, has not been directly and thoroughly addressed in cancer. T cells can sense IFNγ following immunization or infection, leading to the regulation of their differentiation[27–32]. In tumors, although it has been hypothesized that IFNγ induces apoptosis in TILs following checkpoint blockade[21], how chronic IFNγ signaling affects TIL functions and differentiation has not been defined.

In this study, we provide evidence that IFNγ directly acts on TILs to restrict anti-tumor responses, independent of other checkpoints. Characterization of IFNγ receptor (IFNγR) expression in CD8 T cells revealed an intricate regulation of IFNγR relative to T cell subsets in blood of patients with metastatic melanoma. Importantly, IFNγR2 expression negatively correlates with the size of T cell clones and with patients' response to checkpoint blockade, suggesting that CD8 T cells down-regulate their IFNγR for successful anti-tumor immunity. Consistent with this, ablation of the IFNγR specifically in mature CD8 T cells results in greater control of tumor growth in a syngeneic mouse model of melanoma. IFNγR-deficient CD8 T cells display higher expansion and enhanced location at the tumor core, but their fitness is surprisingly not significantly enhanced at the tumor site. Characterization of control and IFNγR-deficient CD8 T cell subsets in tumors confirmed that IFNγ limits TIL expansion. Importantly, our data demonstrate that IFNγ inhibits de novo TIL replenishment by preferentially targeting stem-like T cells. In addition, IFNγ inhibits their maintenance and downstream cytokinesis upon restimulation, which we argue decreases the size of pool of stem-like T cells in tumors over time. As a result, stem-like T-cell clonal diversity is decreased. This provides evidence that IFNγ is involved in a negative feedback loop in melanoma to limit anti-tumor immunity by restricting stem-like T cell longevity, offering new strategies to enhance stem-like T cell potential in tumors and potentially improve adoptive T cell transfer therapies.

## Results

### Regulation of IFNγR expression on circulating CD8 T cells following checkpoint blockade in patients with metastatic melanoma

To characterize the relevance of IFNγ sensing by CD8 T cells during carcinogenesis, we first examined IFNγR expression on T cell subsets in the blood of eight patients with metastatic melanoma before and 21 days after undergoing checkpoint blockade (anti-CTLA-4 and/or anti-PD-1 immunotherapy)[10].

The IFNγR is composed of two subunits; IFNγR1, which contains extracellular domains required for IFNγ binding, and IFNγR2, which is indispensable for signal transduction[33]. CD8 T cells with an activated/ memory phenotype upregulated IFNγR1 and downregulated IFNγR2 compared to naïve CD8 T cells, which has been described for multiple other conditions[34–36]. In fact, most T cells that rapidly produced IFNγ had a similar expression pattern, including MAIT and γδ T cells (Fig. 1a, b). IFNγR2 downregulation suggested that IFNγ-producing cells became less sensitive to IFNγ compared to naïve cells, while increased IFNγR1 expression could reflect the existence of an autocrine regulatory loop where it might be important for CD8 T cells to still sense the IFNγ they produce. Checkpoint blockade did not affect IFNγR expression across those T cell subsets. Mitotic cells are the only subset affected by checkpoint blockade, with decreased proportion of cells expressing IFNγR1 and IFNγR2 transcripts after treatment (Fig. 1a, b). Interestingly, mitotic cells constitute the only subset to exhibit very low levels of both IFNγR chains (Fig. 1a, b), suggesting an escape from an IFNγ-driven regulatory loop.

The presence of large circulating CD8 T cell clones, defined as representing over 0.5% of the blood T cell repertoire, has been shown to associate with good prognosis 6 months after checkpoint blockade[9,10]. Larger clones expressed significantly less IFNγR2 transcripts, whereas no correlation was observed between the expression of IFNγR1 and clonal size (Fig. 1c, d). In addition, checkpoint blockade further decreased the proportion of cells with detectable IFNγR2 expression (Fig. 1d).

Inference of T cell subsets and clonal size from bulk RNAseq yielded similar trends regarding the regulation of IFNγR expression, with low expression of IFNγR2 in subsets that rapidly produce IFNγ (Fig. S1a) and in large clones (Fig. S1b). We characterized the potential consequences of IFNγR dynamic expression by analyzing the genes correlated with expression of IFNγR1 and IFNγR2 (Supplementary Dataset 1 and 2) from bulk RNAseq data. Pathway analysis revealed that both chains are inversely correlated with pathways linked to cytokinesis and migration (Fig. 1e, f). In addition, consistent with the fact that IFNγR1 and IFNγR2 are differentially expressed on different subsets, they associated with distinct pathways. For example, IFNγR2 inversely correlated with pathways related to cytotoxicity (Fig. 1f), in agreement with the fact that IFNγR2 was downregulated after checkpoint blockade and on large clones, as both conditions are characterized by increased cytotoxicity[10].

Importantly, further characterization of bulk RNAseq data from blood CD8 T cells after the first cycle of checkpoint blockade revealed that expression of IFNγR2, but not IFNγR1, was associated with poor response to checkpoint blockade (Fig. 1g), which is consistent with the downregulation of IFNγR2 observed with large clonal size (Figs. 1d and S1b). Interestingly, down-regulation of both IFNγR1 and IFNγR2 significantly correlated with auto-immune adverse effect following checkpoint blockade (Fig. 1h), which indicates that IFNγ sensing in CD8 T cells might regulate tolerance during active immune responses.

Overall, our data indicate that IFNγR expression in CD8 T cells is tightly regulated and negatively correlates with response to immunotherapy in a cohort of metastatic melanoma patients. This led us to hypothesize that IFNγ sensing by CD8 T cells might induce a negative feedback loop impacting their anti-tumor function.

### Ablation of IFNγR in CD8 T cells is enough to enhance tumor control, to improve TIL expansion and intra-tumoral location without substantially increasing TIL fitness

To understand whether IFNγ sensing by CD8 T cells directly affected the efficacy of anti-tumor responses, we used a mouse model whereby IFNγ sensing was ablated in CD8 T cells by deleting IFNγR1 specifically in mature CD8 T cells (Fig. S2a–d, CD8-IFNγRKO). We engrafted those mice with the melanoma cell line B16F10 expressing ovalbumin (B16-OVA) and followed tumor growth over time. Ablation of IFNγR in CD8 T cells resulted in increased overall survival (Fig. 2a) and decreased tumor size (Fig. 2b). Tumor control was observed in 50% of CD8-IFNγRKO mice compared to 20% in control mice (Fig. 2c). Enhanced

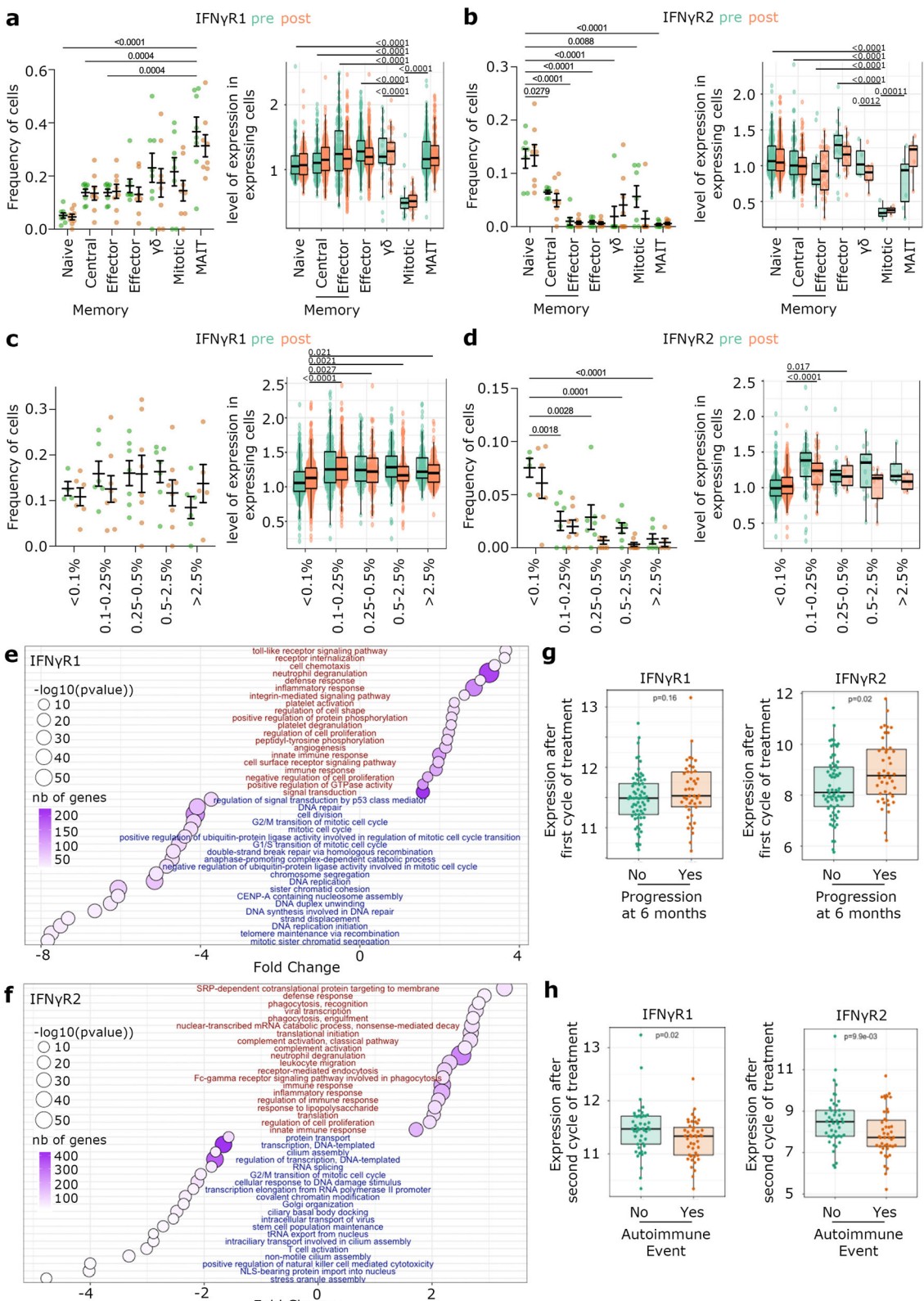

tumor control in CD8-IFNγRKO compared to control mice led to decreased tumor weight (Fig. 2d), which could also be recapitulated with the MC38 cell line (Fig. S2e–f). Decreased tumor growth in CD8-IFNγRKO was associated with increased number of CD8 T cells in tumors (Fig. 2e), regardless of tumor volume (Fig. 2f).

To assess whether increased tumor control was due to enhanced effector functions, we quantified the capacity of TILs from control and

CD8-IFNγRKO mice to secrete pro-inflammatory cytokines. To do so, intra-tumoral CD8 T cells from B16-OVA bearing control and CD8-IFNγRKO mice were isolated and re-stimulated in vitro to quantify IFNγ and TNFα secretion by flow cytometry. Surprisingly, we could not find any statistically significant difference in cytokine secretion (Fig. 2g). Consistent with this, IFNγ concentration in the tumor milieu was similar between control and CD8-IFNγRKO mice, as measured by ELISA

**Fig. 1 | IFNγ receptor expression in circulating CD8 T cells from metastatic melanoma patients undergoing checkpoint therapy and correlation with progression. a–d** Analysis of IFNγR expression in scRNAseq data of CD8 T cells from 8 metastatic melanoma cancer patients treated with checkpoint blockade (pembrolizumab ($n = 4$), ipilimumab/nivolumab ($n = 4$)). Samples were taken before (Day 0, pre) and after (Day 21, post) treatment. **a–b** Frequency of cells displaying detectable expression of the receptor (left panels) and level of expression (right panels) of IFNγR1 (**a**) and IFNγR2 (**b**) in multiple T cell subsets. **c–d** Proportion of cells displaying detectable expression of the receptor (left panels) and level of expression (right panels) of IFNγR1 (**c**) and IFNγR2 (**d**) according to clonal size. For (**c**, right panel and **d**), statistics compare smallest clonal size to others. **a–d** Percent of cells is showed as Mean +/− SEM. Level of expression is showed as box-plot with lower and upper hinges of boxes represent 25th to 75th percentiles, the central line represents the median, and the whiskers extend to highest and lowest values no greater than 1.5× interquartile range. Pairwise group comparisons with two-sided Wilcoxon signed-rank test. **e–h** Bulk-RNA seq analysis of CD8 T cells from metastatic melanoma patients ($n = 110$) 21 days after starting checkpoint blockade therapy. **e–f** GOBP pathway analysis of genes correlated (blue) and inversely correlated (red) with IFNγR1 (**e**) and IFNγR2 (**f**) after checkpoint blockade. Overlap (n), number of significant genes from a pathway (hypergeometric test). **g**- Correlation between disease progression and IFNγR1 (left panels) or IFNγR2 (right panels) expression after first cycle of checkpoint blockade. $N = 68$ no progression; $n = 44$ **h**- Correlation between autoimmune event and IFNγR1 (left panels) or IFNγR2 (right panels) expression after first cycle of checkpoint blockade. $N = 46$ with autoimmunity; $n = 44$ (**g–h**) Lower and upper hinges of boxes represent 25th to 75th percentiles, the central line represents the median, and the whiskers extend to highest and lowest values no greater than 1.5× interquartile range. Two-sided Wilcoxon signed-rank test.

(Fig. 2h). Similarly, we could not detect significant differences in LAMP-1 surface expression, between control and CD8-IFNγRKO TILs following ex vivo restimulation (Fig. 2i), suggesting that IFNγ signaling in CD8 T cells does not impact cytotoxicity. This overall suggested that increased tumor control in CD8-IFNγRKO mice was not primarily due to increased effector functions.

While CD8-IFNγRKO mice display increased CD8 T cell intratumoral expansion (Fig. 2e–f), this is often not sufficient to predict tumor immunity, their localization within the tumor environment being also vitally important[37]. Tumors can be currently classified as one of three major immune infiltration profiles[38,39]: i) *cold* when no immune infiltrate is observed; ii) *hot* when the tumor is infiltrated and inflamed and iii) *excluded*, characterized by T cells localized at the margin with poor infiltration into the tumor core. Excluded phenotype is associated with various epithelial cancers such as melanomas[40] and correlates with poor response to immunotherapy[39]. To investigate whether IFNγR ablation affected TIL location, B16-OVA tumors from control or CD8-IFNγRKO mice were stained for CD8 T cells and their localization within tumors was quantified. Tumors from control mice had an excluded phenotype, with most TILs localized at the margin or trapped within the stromal network, and only 30% were present in the tumor core (Fig. 2j, k). By contrast, TILs from CD8-IFNγRKO mice were more infiltrated, with approximately 50% of CD8 T cells in direct proximity of tumor cells (Fig. 2j, k), suggestive of a hot phenotype. To characterize whether this differential location could be attributed to a function of IFNγ on TIL migratory behavior, we performed in vivo homing assays. In vitro activated WT and IFNγRKO antigen (OVA)-specific CD8 T cells (OTI) were injected in B16-OVA-bearing mice, and their recruitment to tumors was assessed after 24 h. Surprisingly, we found that IFNγRKO OTI had a strong defect in homing to tumors (Fig. S2g). Importantly, it highlighted the fact that increased intratumoral CD8 T cell expansion in CD8-IFNγRKO mice could not be explained by increased homing and suggested the mechanism by which CD8 T cell expanded was dominant and compensated for the lack of homing.

Altogether, our data demonstrate that IFNγR ablation in CD8 T cells improves anti-tumor immunity. In addition, IFNγR ablation in CD8 T cells does not substantially enhance intrinsic functional properties but it increases CD8 T cell expansion and localization to the core of the tumor, despite inhibiting T cell homing. As increased tumor control occurred without the addition of checkpoint blockade, we conclude that IFNγ sensing by CD8 T cells is an independent pathway restricting T cell anti-tumor immunity.

**IFNγR ablation induces the accumulation of intra-tumoral stem-like CD8 T cells, resulting in increased clonal diversity**

To gain further insight into the role of IFNγ on CD8 T cells, CD8 T cells from B16-OVA tumors and draining LNs of control and CD8-IFNγRKO mice were harvested between 12- and 15-days post-tumor engraftment, sorted (Fig. S3a) and subjected to single-cell RNA sequencing (scRNAseq). To understand whether IFNγ acted preliminarily in LN or at the tumor site, we performed Differential Gene Expression analysis between control and CD8-IFNγRKO cells from LN and tumors. Interestingly, we could not detect any transcript differentially regulated in LN, all differences were observed in TILs (Fig. S3b–d). We therefore focused our analysis on tumor samples, where most dysregulation occurred. Using unsupervised hierarchical clustering, we identified 5 clusters (Fig. 3a), all of which expressing multiple exhaustion molecules to varying degrees (Fig. 3b). We observed overall very little diversity between clusters. Cluster 4 had a specific Stem-like signature (Fig. 3c), characterized by the expression of *Tcf1/7*, *IL7r*, *Ccr7*, low expression of cytotoxic molecules such as *Ifng* and *Gzmb* and low expression of transcripts coding for exhaustion molecules such as *Pdcd-1* (PD-1) compared to the other clusters (Fig. 3d). Pseudotime analysis demonstrated that the stem-like cluster gave rise to two distinct trajectories (Fig. 3e); a primary trajectory leading to clusters 0, 1, and 2, which displayed the highest exhaustion scores (Fig. 3f, g), and a secondary trajectory leading to a low-to-intermediate exhaustion scoring cluster 3 (Fig. 3f). Interestingly, cluster 3 also had significantly reduced TCR signaling score (Fig. 3h) and minimal expression transcription factor *Tox* and *Entpd1* (encoding CD39) (Fig. 3i). These overall characteristics might be suggestive of bystander T cells[41–44], for which the TCR is not triggered.

To assess whether IFNγR ablation broadly affected the proportion or differentiation of TILs, we grouped the different clusters in 3 main cell states: Stem-like (cluster 4), Exhausted (clusters 0, 1, 2) and Bystander-like (Byst-like, cluster 3) (Fig. 4a). Quantification of the number of TILs in each "cell state" revealed that TILs from CD8-IFNγRKO mice contained an increased proportion of cells belonging to the Stem-like cluster (Fig. 4b), suggesting that IFNγR deletion specifically induced the accumulation of Stem-like T cells. Indeed, we found a greater percentage of CD8+ PD-1+ T cells that were Tcf7 positive in tumors from CD8-IFNγRKO compared to control mice in vivo (Fig. 4c). A similar result was observed when examined LN samples, suggesting that IFNγ affected cell type proportions mainly at the tumor site (Fig. S3e, h). Consistent with Stem-like T cells being an important target of IFNγ for inhibiting long-term anti-tumor responses, WT stem-like T cells displayed higher IFNγR expression compared to all other intra-tumoral CD8 T cell subsets (Fig. 4d). This resulted in enhanced responsiveness to IFNγ in tumors, as exemplified by increased expression of multiple IFNγ-responsive genes compared to other T cell subsets (Fig. 4e). This suggested that stem-like T cells had an enhanced sensitivity to IFNγ.

Stem-like T cells are continuously depleted at the tumor site[15,16], which might result in their intra-tumoral clonal diversity being limited. We reasoned that the increased stem-cell maintenance in CD8-IFNγRKO mice would lead to an accumulation of stem-like T cell clones, thereby enhancing stem-like T cell diversity. To explore clonal landscape, we performed coupled transcriptome and TCR single-cell sequencing of TILs from control and CD8-IFNγRKO mice. While there

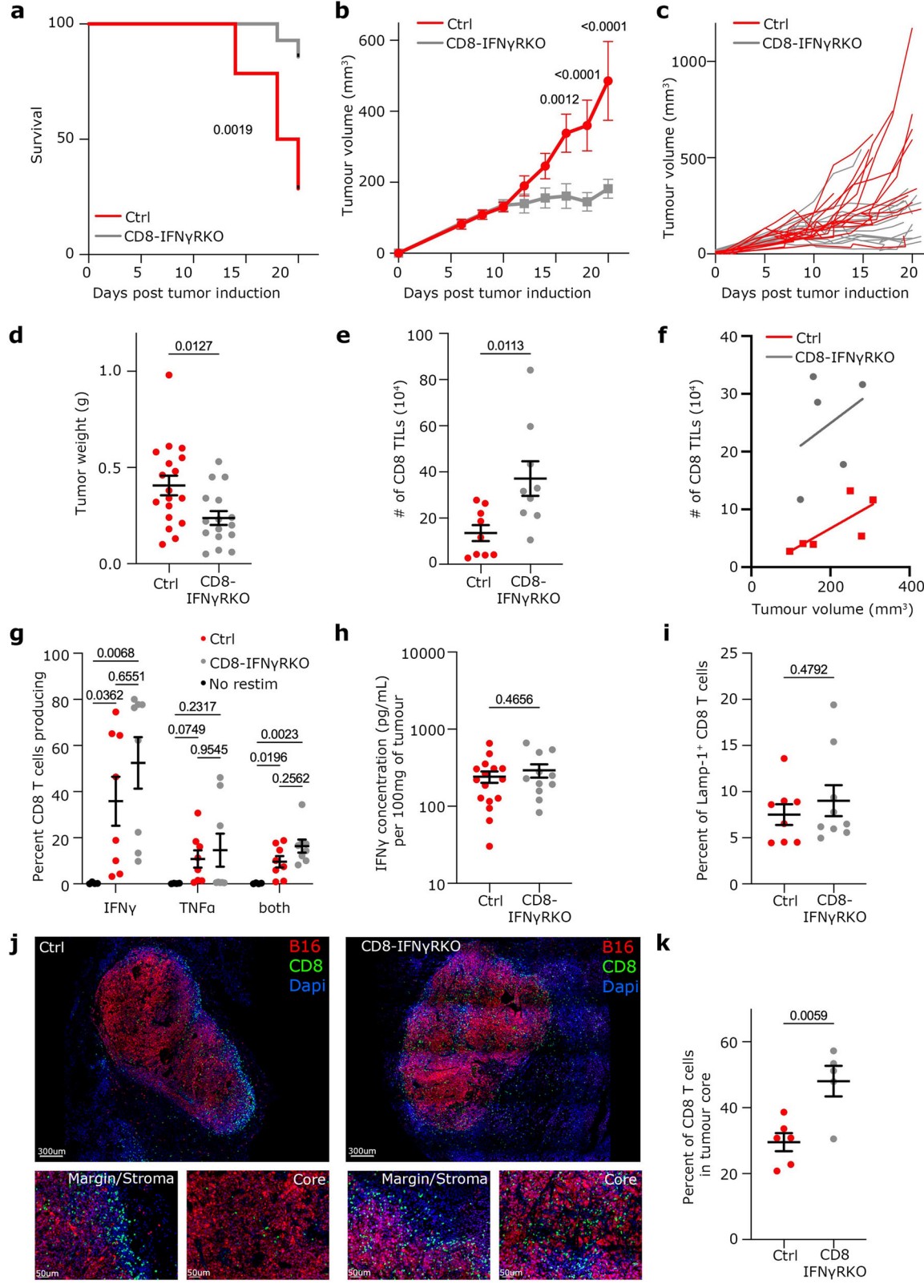

was no clonal overlap between mice (Fig. S4a, b), analysis of the top 20 clones revealed a high degree of overlap between Stem-like T cells and the other T cell subsets for both control and CD8-IFNγRKO mice (Figs. 4f and S4b), indicating that most exhausted T cells originated from stem-like T cells and confirming the relevance of stem-like T cells in this model. Bystander-like T cells seemed to share the same TCRs as exhausted T cells and stem-like T cells, regardless of the genotype

(Figs. 4f and S4b), however, because we recovered about 15% of clonotypes in the Byst-like T cell state (Fig. S4c), it was unclear whether this subset represented antigen-specific TILs that lost TCR reactivity or true bystanders. Analysis of the relative abundance of T cell clones in control and CD8-IFNγRKO TILs revealed that Stem-like T cells displayed the highest diversity (Fig. S4d) and were characterized by low-abundance clones (Fig. 4g). Interestingly, ablating the IFNγR resulted

**Fig. 2 | IFNγ sensing by CD8 T cells restricts anti-tumor immunity in mice.**
**a–f** Control (red) and CD8-IFNγRKO (grey) mice were engrafted with B16-OVA tumors. **a**- Mouse survival over time (*n* = 14). Data is from 3 independent experiments. Mantel-Cox test. **b**- Average tumor volume over time. Data is from 3 independent experiments (*n* = 14). Data shows Mean +/− SEM. Mixed-effects model analysis with Šidák's multiple comparison test. **c**- Tumor volume of individual mice over time. **d**- Tumor weight between 12 and 15 days. Each point represents one mouse (*n* = 18 Control; *n* = 16 CD8-IFNγRKO), from 3 independent experiments. Data shows Mean +/− SEM. Unpaired *t* test. **e–f** Analysis of total number of TILs per tumor (**e**, *n* = 9) and the number of TILs according to the tumor volume (**f**, *n* = 6 Control and *n* = 5 CD8-IFNγRKO) after 12 to 15 days. Each point represents one individual mouse from 3 independent experiments. Data shows Mean +/− SEM. Unpaired *t* test. **g**- TILs were harvested and restimulated in vitro with 1ug OVA peptide. Quantification of IFNγ and TNFα producing cells. Every dot is a mouse (*n* = 4 Naïve control; *n* = 8 WT/ CD8-IFNγRKO) from 2 independent experiments. Data shows Mean +/− SEM. Two-way Anova with Šidák's multiple comparison test.

**h**- IFNγ quantification by Elisa in tumor supernatants from control and CD8-IFNγRKO mice after 12 to 15 days. Every dot is a mouse *(n* = 16 Control; *n* = 11 CD8-IFNγRKO) from 2 independent experiments. Data shows Mean +/− SEM. Unpaired *t* test. **i**- TILs were restimulated in vitro with 1ug OVA peptide. Quantification of surface LAMP-1 staining. Each symbol represents an individual mouse (*n* = 8 Control; and *n* = 9 CD8-IFNγRKO) from 2 independent experiments. Data shows Mean +/− SEM. Unpaired *t* test. **j–k** B16-OVA tumor sections from Control and CD8-IFNγRKO mice were stained with for CD8 (green). Tumors express mCherry (red) and nuclei were stained with Dapi (blue). The tumor core was defined as mCherry+ Dapi+. The stroma network and margin were defined as mCherry- Dapi+. **j**- Representative images of tumor sections from control (left panels) and CD8-IFNγRKO (right panels) mice. **k**- Proportion of CD8 T cells in the core of tumors from Control (red) and CD8-IFNγRKO (grey) mice. Every dot is a mouse (*n* = 6 Control; *n* = 5 CD8-IFNγRKO) from 3 independent experiments. Data shows Mean +/− SEM. Unpaired *t* test.

---

in an accumulation of rare clones (Fig. 4g). This translated into an increased abundance of smaller clones in the Exhausted state following IFNγRKO ablation, as defined by representing less than 0.05% of the repertoire. Plotting the relative abundance of T cell clones on the UMAP corroborated the fact that the Stem-like T cell cluster mainly contained low-abundance clones (Fig. 4h). In addition, this also indicated that, while control TILs were mainly composed of larger clones, CD8-IFNγRKO had a breadth of clonal sizes found throughout the path of exhaustion (Fig. 4g, h).

To conclude, IFNγR deletion in CD8 T cells results in a greater proportion of stem-like T cells in the tumor, leading to increased stem-like T cells clonal diversity. This suggests that IFNγ signaling in intra-tumoral CD8 T cells inhibits stem-like CD8 T cells generation or maintenance.

## IFNγ has a widespread effect on T cell proliferation which originates from stem-like T cells

Because we observed a greater expansion of IFNγR deficient CD8 T cells and IFNγ is known to induce cell cycle arrest[45], we investigated whether IFNγR deletion increased their proliferation. We first quantified the proportion of control and IFNγRKO CD8 T cells in the different cell cycle phases based on transcriptomics data. Cell cycle scoring revealed that 61% of intra-tumoral CD8 T cells from CD8-IFNγRKO mice were in G2/M or S phase compared to 38.9% in control mice (Fig. 5a). Differential pathway analysis between CD8-IFNγRKO and control CD8 T cells within the different cell states revealed an increase of Normalized Enrichment Score (NES) for multiple pathways associated with cell cycle, this for all cell states (Figs. 5b, c and S5a, b). This is consistent with the fact that pathways related to cell cycle are inversely correlated with IFNγR expression in metastatic melanoma patients (Fig. 1e–f). IFNγ-induced cell cycle arrest was confirmed by investigating the proliferative capacity of antigen-specific CD8 T cells in the presence or absence of IFNγ sensing. To do so, control and IFNγRKO OVA-specific CD8 OTI T cells stained with a proliferation dye were transferred in B16-OVA bearing mice and proliferation was assessed by quantifying proliferation dye dilution. As expected, IFNγRKO OTI T cells displayed increased proliferation compared to control OTI (Fig. S5c, d). These data overall indicated that ablation of IFNγ sensing resulted in an overall increased in CD8 T cell expansion due to increased proliferative capacity.

To get further insight into relationship with cell states during proliferation, we isolated all tumor-infiltrating immune cells from WT mice, treated those with anti-CD3 and anti-CD28, and assessed how IFNγ sensing affected stem-like and exhausted T cells proportion ex vivo. To do so, we quantified percentage of dividing cells that expressed Tcf1/7 as a read-out of stem-like T cells following restimulation in the presence or absence of recombinant IFNγ or blocking IFNγ antibodies. Analysis of Tcf1/7 expression during proliferation revealed that Tcf1/7+ T cells were maintained throughout multiple cycles of

division (Fig. S5e), confirming that those cells indeed had stem-like properties. Anti-IFNγ treatment resulted in increased proportion of PD-1+ Tcf1/7+ stem-like T cells compared to control cells or cells treated with recombinant IFNγ (Fig. 5d). To validate the hypothesis that anti-IFNγ treatment resulted in an accumulation of Stem-like T cells as opposed to a diminution of other cells, we analyzed the actual number of PD-1+ Tcf1/7+ stem-like T cells over time. Anti-IFNγ treatment resulted in increased retention and/or decreased Stem-like T cell disappearance compared to other conditions (Fig. 5e). Supporting this, a greater percentage of Stem-like T cells started proliferating when IFNγ was inhibited (Fig. 5f). These data suggested that IFNγ blocking resulted in the accumulation of stem-like T cells. To confirm this, we sorted stem-like T cells or exhausted T cells from B16-OVA tumors (Fig. S5f) and restimulated them as above. As previously described[13,46], stem-like T cells had enhanced proliferative capacity compared to exhausted T cells (Fig. 5g, h), with 35% of stem-like T cells that completely diluted the proliferation dye, as opposed to 5% for exhausted T cells (Fig. 5i). Importantly, exhausted T cell proliferation was not rescued by anti-IFNγ treatment (Fig. 5j, k), confirming that IFNγ did not directly affect exhausted T cells, but rather targeted stem-like T cells. Because IFNγ did not directly target exhausted T cells but most cells exhibiting transcriptional cell cycle signature belonged to the Exhausted/Bystander states (Fig. S5g), we concluded that cycling cells corresponded to newly differentiated progeny downstream of stem-like T cells. Finally, to confirm that increased intratumoral CD8 T cell expansion observed in CD8-IFNγRKO mice was supported by stem-like T cells as opposed to exhausted T cells, we used lineage cell-tracing strategies. WT and IFNγRKO OTI T cells were differentiated in vitro into stem-like and exhausted T cells following previously defined protocols[47,48] (Fig. 5l, m). WT and IFNγRKO cells were ad-mixed and transferred in B16-OVA tumor-bearing mice. The ratio between KO and WT OTI cells in tumors was analyzed after 6 days. Five times more WT than IFNγRKO exhausted T cells were found in tumors, consistent with the defect of IFNγRKO cells to home to tumors (Fig. S2g). Of note, we found that the expression of multiple chemokine receptors was differentially expressed between control and CD8-IFNγRKO mice (Fig. S5H), potentially explaining the defect in homing induced by IFNγR deletion. Importantly, stem-like T cells had the potential to recover from the defect in homing, with half of the tumors exhibiting more IFNγRKO than WT cells when cells originated from stem-like T cells (Fig. 5n).

Altogether, our data demonstrate that IFNγ sensing by T cells leads to a widespread defect in proliferation which originates from stem-like T cells.

## IFNγ inhibits the self-renewal capacity of intra-tumoral stem-like T cells by inhibiting their maintenance

While increased stem-like T cell proliferation could explain enhanced expansion observed in CD8-IFNγRKO mice, it could not account for

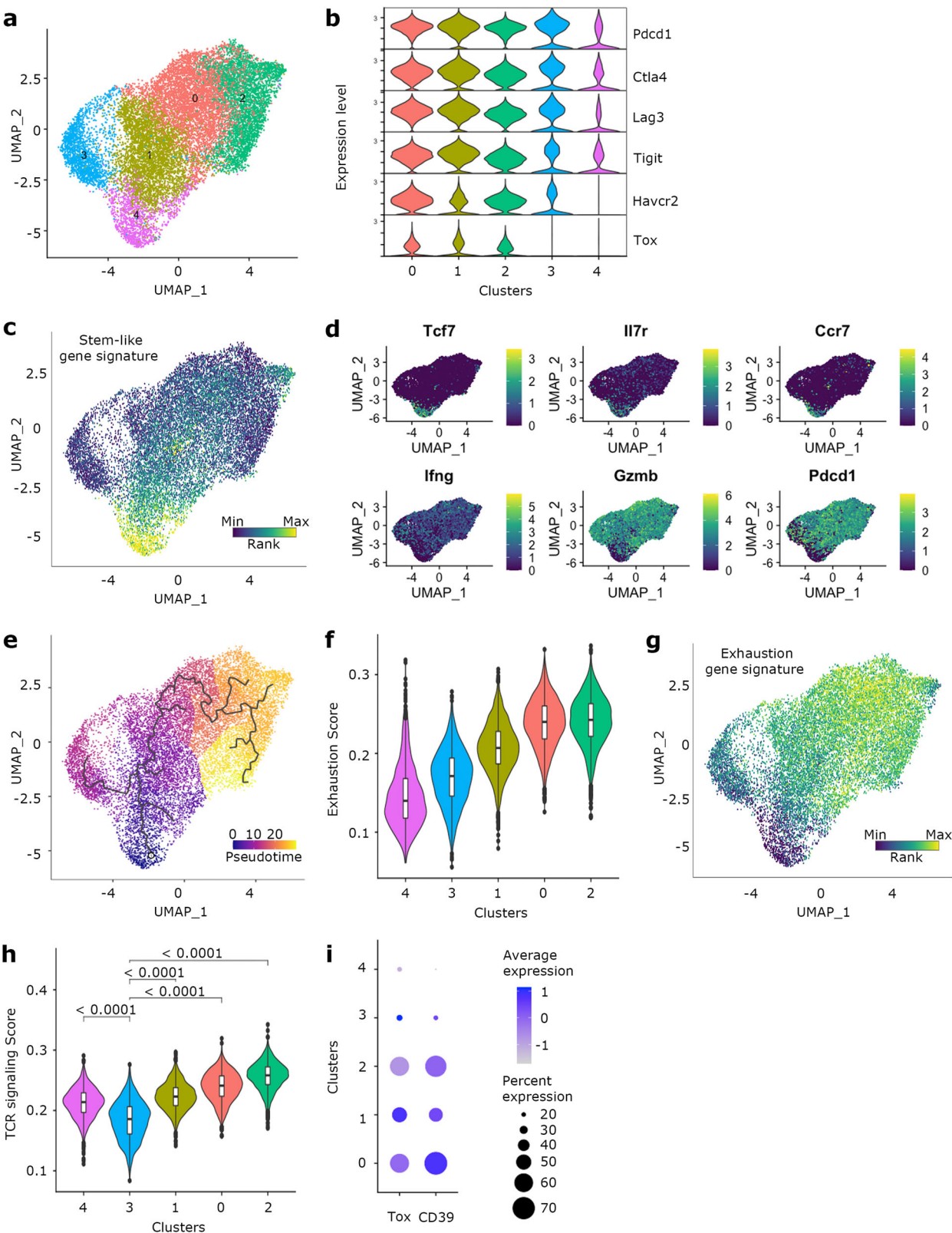

increased clonal diversity, suggesting that IFNγ sensing by stem-like T cells had complex outcomes. To further characterize the relevance of IFNγ signaling in stem-like T cells, we performed Differential Gene Expression and pathway analysis between CD8-IFNγRKO and control cells from the stem-like T cell cluster. Overall, the main increased pathways in CD8-IFNγRKO stem-like T cells compared to control cells were related to Cell cycle, Heat-shock response and DNA repair

(Fig. 6a, b). Altogether, those pathways were reminiscent of increased stemness and resistance to stress, which could suggest enhanced resistance to apoptosis. We analyzed whether IFNγ could induce cell death of intra-tumoral T cell subsets, as suggested in previous studies[21] using caspase 3 cleavage as a readout. Surprisingly, IFNγR deletion did not affect the percentage of apoptotic stem-like or exhausted T cells in vivo (Fig. 6c). Similarly, blocking IFNγ had no effect on apoptosis of

**Fig. 3 | Unbiased characterization of CD8 T cells infiltrating B16-OVA tumors.** Control (n = 3) and CD8-IFNγRKO (n = 3) mice were engrafted with B16-OVA tumors. CD45⁺CD3⁺CD8⁺Tomato⁺ cells were sorted from tumors and subjected to scRNA-seq analysis. **a**- Graph-based clustering (n = 15706 total) identified 5 clusters. **b**- The violin plots show the expression levels (y axes) of selected exhaustion markers (Pdcd1 (encoding PD-1), Ctla4, Lag3, Tigit, Havcr2 (encoding Tim-3), and Tox) in each of the identified clusters (x axes). **c**- UMAP visualization of CD8 Stem-like signature. **d**- UMAP visualization of the distribution of selected transcripts defining the Stem-like gene signature. Genes include Tcf7 (encoding Tcf1), Il7r, Ccr7, IFNγ, Gzmb (encoding granzyme B) and Pdcd1 (encoding PD-1). **e**- Pseudotime trajectory across the 5 TIL clusters. **f**- The violin plots show the expression score (y axes) of the exhaustion gene signature in each of the identified clusters (x axes). Clusters have been re-ordered to follow the exhaustion score and pseudo-time. Box plots indicate median (middle line), 25th, 75th percentile (box). (n = 15706 total). **g**- UMAP visualization of CD8 T cell terminal exhaustion signature. **h**- The violin plots show the expression score (y axes) of the TCR signaling gene signature in each of the identified clusters (x axes). Box plots indicate median (middle line), 25th, 75th percentile (box). Pairwise group comparisons with two-sided Wilcoxon signed-rank test. (n = 15706 total). **i**- The dot plot shows the relative expression and percent of cells expressing the transcription factor Tox and the receptor Entpd1 (encoding CD39) within the different clusters.

in vitro activated OTI stem-like T cells[49] (Fig. 6d). While very few transcripts were differentially regulated between in CD8-IFNγRKO and control stem-like T cells (115 down-regulated; 78 up-regulated in CD8-IFNγRKO, Supplementary Dataset 3), the two most up-regulated transcripts corresponded to *Hspa1a* and *Hspa1b*, two members of Heat Shock Proteins (HSPs) (Fig. 6b). This was not a consequence of IFNγR deletion, as CD8 T cells from LN did not exhibit any transcript that were differentially expressed (Fig. S3c). Levels of HSPs are known to be higher in multiple stem cell types compared to their differentiated counterparts[50,51]. We indeed could find increased expression of transcripts related to heat stress in Stem-like T cells compared to exhausted T cells in our dataset (Fig. S6a), together with decreased apoptosis signature (Fig. S6b), consistent with an active involvement of HSPs in stem-like T cell survival in tumors.

We therefore investigated whether IFNγ decreased stemness potential or maintenance of stem-like T cells. To do so, we first quantified the transcriptional stem-like score using two curated stem-like signatures[52,53] and found that CD8-IFNγRKO stem-like T cells had a higher stem-like signature score compared to controls (Fig. 6e), suggesting that IFNγ indeed interfered with stemness in T cells. To test whether IFNγ inhibited the maintenance of stem-like T cells in vivo, we transferred in vitro generated WT and IFNγRKO OTI stem-like T cells in the same B16-OVA bearing mice and quantified the percentage of OTI cells that still expressed the stemness marker Tcf1/7 after 6 days. IFNγRKO OTI cells had a greater ability to retain the expression of Tcf1/7 (Fig. 6f, g), demonstrating that IFNγ decreases stem-like T cell maintenance. Similar results were obtained when we isolated endogenous stem-like T cells and analyze their maintenance in the presence or absence of anti-IFNγ after restimulation (Fig. 6h).

Because the β-catenin pathway has been shown to promote stemness of T cells[48], and intra-tumoral stem-like T cells from CD8-IFNγRKO mice displayed enhanced Wnt/β-catenin transcriptional signature score compared to control cells (Fig. S6c), we investigated whether IFNγ interfered with the Wnt/β-catenin pathway. Surprisingly, we found that β-catenin expression was increased following TCR stimulation, as previously described[54], but not Wnt (Fig. S6d). In addition, TCR-induced β-catenin was reduced by IFNγ treatment (Fig. S6e), suggesting that IFNγ might inhibit the intrinsic capacity of T cells to maintain a self-renewing pool following activation[49,55]. Consistent with this, TCR-induced generation of stem-like T cells was further potentiated by anti-IFNγ treatment (Fig. S6f). Interestingly, this is no longer observed when stem-like T cells are generated in the presence of the GSK3β inhibitor TWS119 (Fig. S6f) suggesting that an alternate pathway might be at play[56]. Overall, this indicates that IFNγ inhibition reinforces intrinsic T cell stemness.

Finally, to investigate the relationship between IFNγR expression and stemness in human tumors, we analyzed datasets from patients with melanoma and non-small cell lung cancer. We revealed an inverse relationship between the percent of stem-like T cells with a detectable expression of IFNγR1 and stem-like signature (Fig. 6i). This inverse correlation is consistent with a function of IFNγ in inhibiting the maintenance of stem-like T cells. Interestingly, in circulating CD8 T cells from metastatic melanoma patients undergoing checkpoint

blockade, IFNγR2 expression inversely correlated with stemness maintenance pathways (Fig. 1f), supporting the notion that IFNγ signaling inhibited CD8 T cell stemness also in blood, at least upon checkpoint blockade.

Overall, these data indicate that IFNγ signaling in intra-tumoral Stem-like CD8 T cells inhibit their self-renewal potential. It is important to note that in the ex vivo restimulation experimental set-up, all cells had encountered IFNγ intratumorally, indicating that inhibition of self-renewal capacities by IFNγ is reversible.

## Deletion of IFNγ signaling leads to phenotypically less exhausted TILs

Analysis of T cell clonality suggested that the increased self-renewal capacity of stem-like T cells in CD8-IFNγRKO mice would perpetuate the generation of newly exhausted T cells, leading to an overall decreased exhaustion state without affecting the path of exhaustion per se. To characterize the relevance of IFNγ signaling on exhaustion, we analyzed differentially enriched pathways between exhausted TILs from control and CD8-IFNγRKO mice. Most enriched pathways in IFNγR deficient exhausted TILs were related to metabolism (Fig. 7a, b). This was also true for the Bystander-like cell state (Fig. S5a, b). Glycolysis was observed mainly in the Exhausted cell state compared to the other cell states (Fig. S7a). Within the Exhausted cluster, TILs from CD8-IFNγRKO mice displayed increased glycolysis score (Fig. 7c). Long-term antigen exposure has been shown to change the metabolic requirements for T cell function, with exhausted T cells relying on oxidative phosphorylation (Oxphos) and fatty acid oxidation (FAO) to produce ATP for IFNγ synthesis[57]. Indeed, increased Oxphos signature correlated with increased Exhaustion in our dataset (Fig. S7a, b). Both Exhausted and Bystander-like T cells from CD8-IFNγRKO mice displayed an increased Oxphos score compared to control cells (Fig. 7d). Increased metabolism-related transcripts could reflect true changes in metabolism, or rather indicate indirect consequences of activation or environmental adaptation. To characterize whether IFNγ directly affected the metabolism of exhausted T cells, we analyzed mitochondrial function ex vivo. Exhausted WT and IFNγRKO CD8 T cells harbor similar mitochondrial bioenergetics as shown by the relative uptake of mitochondrial dyes[58,59] (Figs. 7e and S7c). We then analyzed the Bioenergetic Cellular (BEC) index, a measure of the dependence on respiratory capacity over glycolysis rates[60]. To do so, we quantified the ratio between β-F1-ATPase (ATPB) and GAPDH and found no difference between control and IFNγRKO CD8 T cells (Fig. 7f). Similarly, both cell types exhibited similar GLUT1-dependent glycolysis (Fig. S7d), suggesting similar ATP production levels through aerobic and anaerobic glycolysis. The tumor microenvironment is hypoxic[61], which can naturally lead to anaerobic glycolysis with the upregulation of LDHA expression in cells. IFNγRKO exhausted CD8 T cells have decreased LDHA expression level (Fig. S7e), suggesting that IFNγRKO CD8 T cells relied less on anaerobic glycolysis. Altogether, we concluded that the transcriptional increase in Oxphos observed in IFNγRKO CD8 T cells reflected environmental adaptation rather than intrinsic enhanced metabolism capacities.

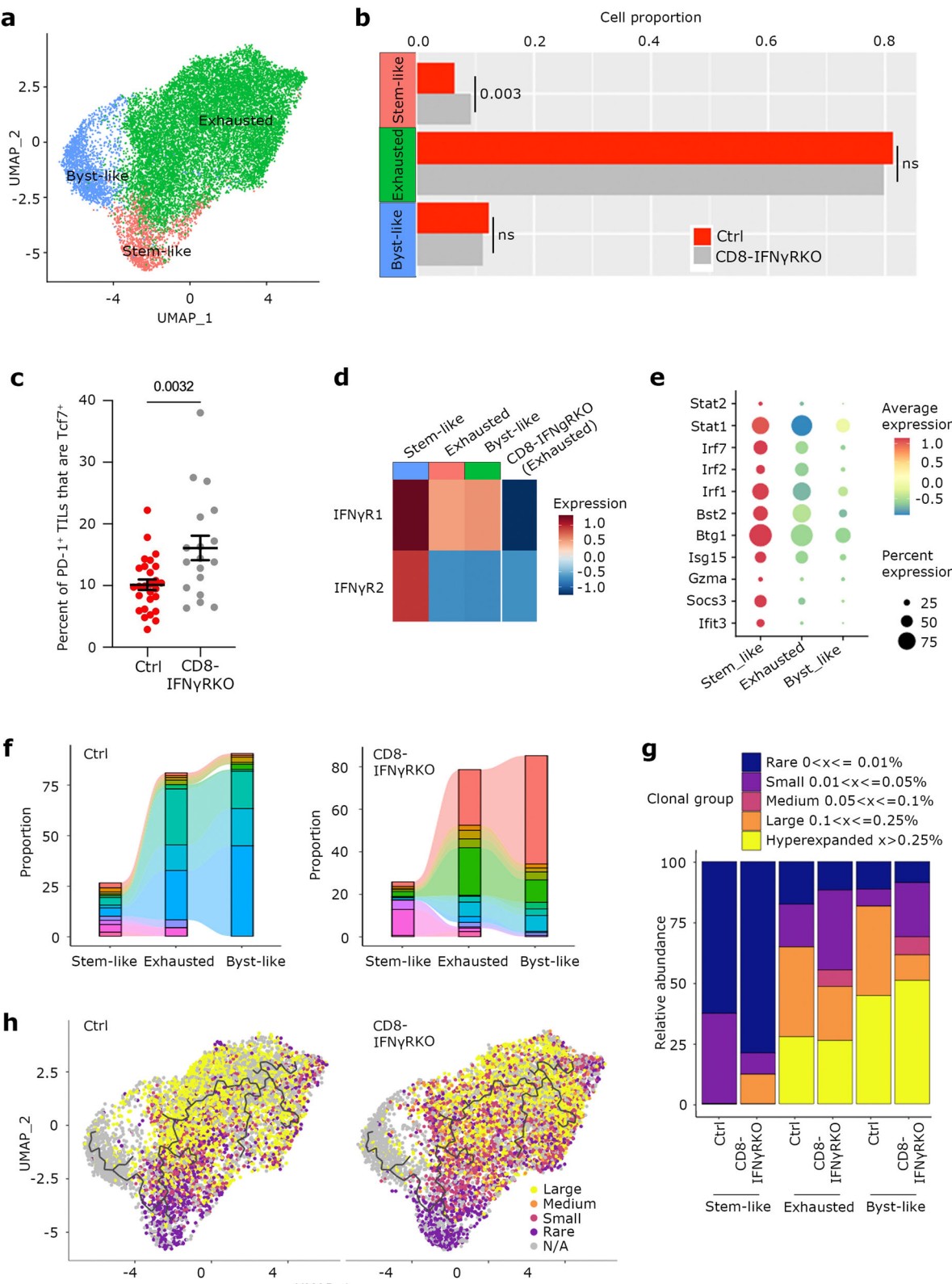

Because exhausted T cells have been shown to become defective in mitochondrial metabolism overtime[62], increased Oxphos signature in CD8-IFNγRKO could also indirectly reflect a lesser level of exhaustion. To address this, we compared the average expression of multiple exhaustion receptors between IFNγRKO and control TILs. Intratumoral IFNγRKO CD8 T cells consistently displayed decreased average expression of multiple exhaustion markers regardless of the cluster they belonged to (Fig. 7g). Focusing on the Exhausted cell state, we could confirm that the reduction in exhaustion markers, albeit slight, was highly significant (Fig. 7h). Differential Pathway analysis confirmed that IFNRKO TILs displayed decreased *Pdcd-1* (PD-1) and Ctla4 (CTLA-4) signaling transcriptional signature compared to control TILs (Fig. 7i). Similar data were observed for Bystander-like TILs (Fig. S7f, g). To confirm that IFNγ influenced the level of T cell

**Fig. 4 | IFNγ sensing by CD8 T cells in tumors alters their cell states and TCR clonality.** Control and CD8-IFNγRKO mice were engrafted with B16-OVA tumors. **a–b** and **d–e** CD45⁺CD3⁺CD8⁺Tomato⁺ cells were sorted from tumors and subjected to scRNA-seq analysis. **a**- Graph-based clustering (*n* = 15,706 total) of the assembled cell states according to the signatures identified in Fig. 4. **b**- The bar plots show the percentages of CD8 T cells from Control (red) and CD8-IFNγRKO (grey) mice that were found in each cell state. Permutation test. **c**- Tumors were harvested 10-12 days post-tumor engraftment and cell suspensions were stained for CD45, CD3, PD-1, and TCF1/7 and subjected to flow cytometry. Percentage of TCF1/7⁺ PD-1⁺ CD8 T cells in tumors from Control (red) and CD8-IFNγRKO (grey) mice Each point represents one individual mouse (*n* = 27 for Ctrl; *n* = 18 for CD8-IFNγRKO), from 4 independent experiments. Data shows Mean +/− SEM. Unpaired *t* test. **d**- Heatmap of control TILs shows the relative average expression of IFNγR1 and IFNγR2 in the different cell states in control mice. The average expression of IFNγR1 and IFNγR2 in CD8-IFNγRKO cells from the exhausted state was used as a control for negative

IFNγR1 expression. **e**- The dot plot shows the relative average expression and percent of cells expressing transcripts implicated in IFNγ signaling and response within the different cell states. **f–h** CD45⁺CD3⁺CD8⁺Tomato⁺ cells were sorted from tumors and subjected to scRNA-seq and scTCR-seq analysis. **f**- The relative frequency of the top 20 clones and their distribution throughout the cell states was analyzed within Control and CD8-IFNγRKO mice. Each color corresponds to a different clone. **g**- Relative TCR clonal abundance by cell state and genotype. Clonotype abundance (as frequency of total repertoire per cell state per genotype) is defined as: Rare = 0 < X <= 0.01%; Small = 0.01 < X <= 0.05%; Medium = 0.005 < X <= 0.1%; Large 0.1 < X <= 0.25%; Hyperexpanded 0.25 < X <= 1%. **h**- Graph-based clustering of TILs (*n* = 7000 for each genotype) from control (left) and CD8-IFNγRKO (right) mice overlaid with the frequency of clonotypes. The black line corresponds to the pseudotime trajectory, as in Fig. 3e. Clonotype abundance is defined as in (**g**).

exhaustion, we performed multidimensional flow cytometry to characterize the expression level of multiple exhaustion markers. We focused our analysis on exhausted PD-1⁺ Lag3⁺ T cells. We identified a subpopulation of exhausted T cells that exhibited a lower expression of multiple exhaustion markers (PD-1, Tim-3, Lag-3 and Tigit) (Fig. 7j), which we referred to as intermediate. Interestingly, exhausted CD8-IFNγRKO T cells contained an increased percentage of the intermediate population (Fig. 7k, m), consistent with a lesser level of exhaustion. Similarly, when we injected in vitro generated stem-like WT and IFNγRKO OTI cells in the same B16-OVA bearing mouse, as in Fig. 6, OTI stem-like T cells converted to exhausted T cells (Fig. S7h). Among those newly differentiated cells, we found a higher proportion of intermediate cells within the exhausted pool after 6 days (Fig. S7i), confirming that the intermediate population originates from stem-like T cells and that IFNγ sensing by stem-like T cells leads to a diminution of this population. Overall, these data are consistent with a steady generation of TILs in CD8-IFNγRKO compared to control mice. This could also explain the subtle increase in cytokine production observed experimentally in IFNγRKO CD8 T cells (Fig. 2) and suggests it could be driven by a sub-population of TILs that recently got generated.

To conclude, our data demonstrate that IFNγ inhibits the self-renewal maintenance of stem-like T cells. Interfering with this mechanism preserves stem-like T cell maintenance, driving the steady generation of newly exhausted T cells with enhanced functional potential, intra-tumoral invasion and proliferative capacity. This overall leads to improved anti-tumor immunity.

## IFNγR deletion improves tumor control following T cell transfer therapy

Adoptive cell therapy (ACT) is a form of treatment that uses the patient's own T cells to eliminate cancer. ACT has shown spectacular success with blood cancers[63], but its use in solid cancer remains limited. Given our findings that IFNγ signaling in CD8 T cells restricts their expansion and stem-like T cell longevity, we hypothesized that deleting the IFNγR in CD8 T cells could improve ACT. To test this, we engrafted WT mice with B16-OVA tumor cells and adoptively transferred either WT or IFNγRKO OTI CD8 T cells when tumors were palpable. Tumor growth was followed over time. Transfer of IFNγ-deficient OTI overall enhanced mouse survival (Fig. 8a), slowed tumor growth (Fig. 8b) and decreased tumor weight (Fig. 8c), demonstrating that IFNγR ablation in adoptively transferred T cells improved tumor control. To address whether IFNγR ablation also affected T cell expansion in this model, we co-transferred WT and IFNγRKO OTI cells in tumor-bearing mice at a ratio of 1:1. Quantification of the ratio between IFNγRKO and WT OTI cells 5 to 7 days after OTI transfer revealed that IFNγR ablation in OTI cells increased OTI expansion 2 to 3-fold compared with WT OTI, both in the tumor and the draining lymph nodes (Fig. 8d). This was overall not related to differences in activation. Indeed, WT and IFNγRKO OTI cells had similar priming

potential, as assessed by CD69 up-regulation following activation with Dendritic cells presenting different altered OVA peptides (Fig. 8e). In addition, we assessed the relevance of IFNγ signaling for OTI functional fitness. WT and IFNγRKO OTI cells were co-transferred in B16-OVA bearing mice and tumors were harvested 7 days after OTI cell transfer. Cytokine secretion was assessed following in vitro restimulation with the OVA peptide SIINFEKL. IFNγR ablation resulted in a mild but consistent increase in IFNγ and TNFα expression (Fig. 8f, g). We did not observe any significant difference between WT and IFNγRKO OTI cells when we performed cytotoxic assays (Fig. 8h, i) and extra-cellular LAMP-1 staining (Fig. 8j), indicating that OTI T cells had a similar cytotoxic capacity regardless of IFNγR ablation.

Our data demonstrate that while IFNγR ablation does not enhance T cell cytotoxic functions, it nevertheless improves T cell expansion and anti-tumor potential in a T cell transfer model.

Altogether, we provide evidence that IFNγ signaling in CD8 T cells inhibits anti-tumor responses by limiting T cell expansion and infiltration. Mechanistically, IFNγ restricts stem-like T cells by limiting their maintenance and inhibiting downstream cytokinesis upon restimulation. This leads to a decrease in Stem-like T cell diversity and a widespread effect on exhausted T cells by reinforcing their dysfunction. Overall, chronic IFNγ signaling in CD8 T cells triggers a negative feedback loop designed to restrict the immune response. We speculate that this is a physiological regulatory mechanism emerging from collective responses[64] to preserve tolerance and prevent immuno-pathology. This is in agreement with the correlation between autoimmune-related adverse events following checkpoint blockade and low expression of IFNγR in CD8 T cells of metastatic melanoma patients (Fig. 1h).

## Discussion
CD8 T cells are critical to the elimination of tumor cells; however, they are often restricted in their cytotoxic capabilities due to immunosuppressive factors. As such, T cell tolerance and dysfunction are major hurdles for successful immune therapies[65,66]. T cell dysfunction, or exhaustion, is a dynamic process, which ultimately leads to a permanent loss of effector functions. Therefore, terminally exhausted T cells cannot be harnessed by immunotherapies[67]. Indeed, checkpoint blockade has been shown to target T cells that have just entered tumors[68] and exhibit a stem-like phenotype[11–14,69]. Stem-like T cells are therefore obvious targets for new immune therapeutic avenues, and there is a critical need to understand how this population is generated and maintained[70]. In this study, we contribute to this important question by describing a role for IFNγ in regulating stem-like T cell maintenance intratumorally.

While the generation and maintenance of stem-like T cells in LNs has received increasing attention[15,16,71], it remains unclear why they are not maintained at the tumor site. It has been proposed that loss of antigen presentation to stem-like CD8 T cells, required to continuously

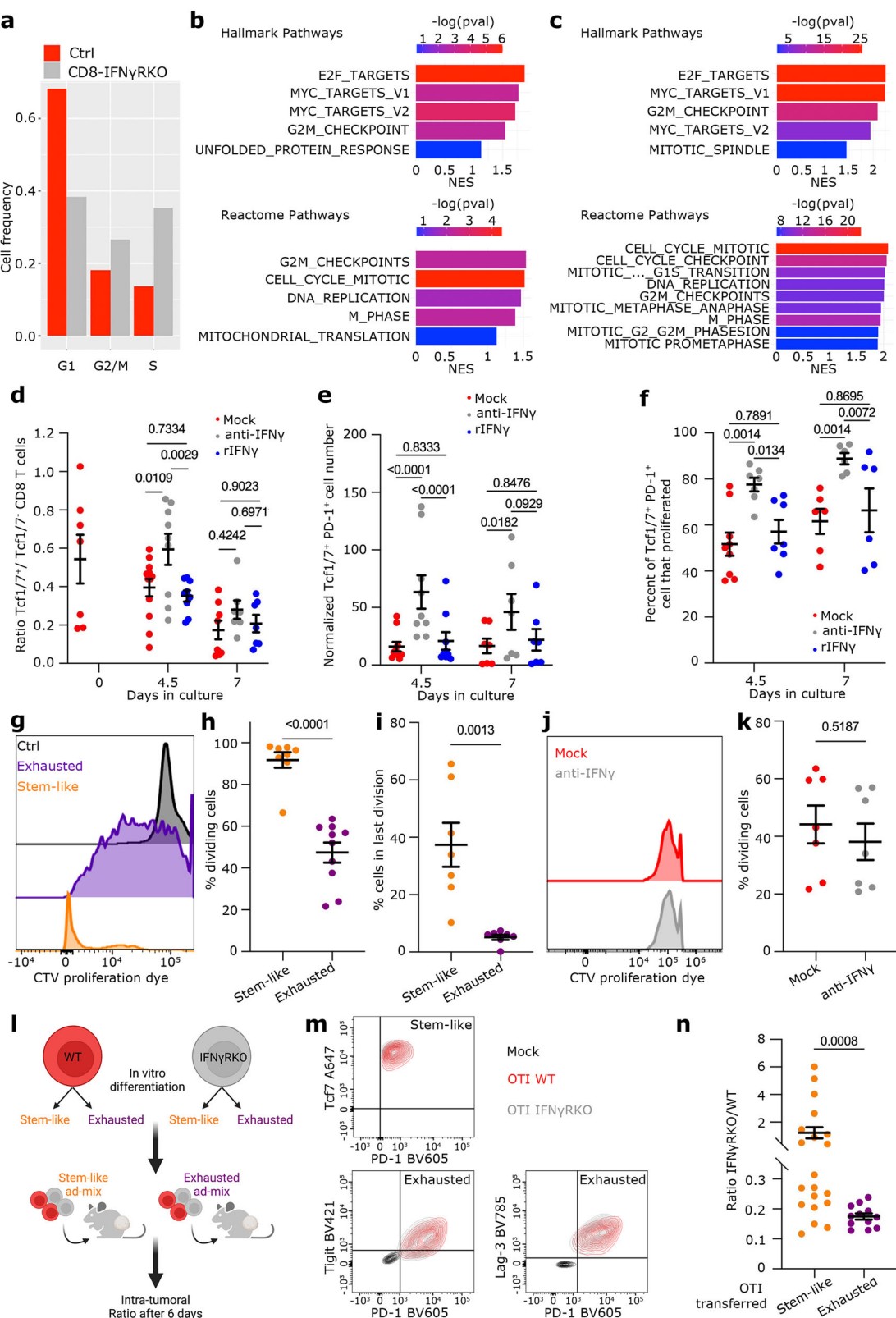

produce terminally differentiated CD8 T cells, causes the decline in T cell responses in cancer, rather than expression of checkpoint molecules[46]. Nevertheless, it has also been suggested that chronic antigen presentation ultimately depletes stem-like T cells via Tcf1/7 down-regulation[15,72]. Our work provides insight into stem-like T cell regulation at the tumor site by providing a mechanism whereby IFNγ sensing induces the loss of stem-like T cells. By contrast, IL10 sensing

has been recently correlated with stem-like T cell maintenance in chronic lymphocytic leukemia[73], indicating that the balance between pro- and anti-inflammatory cytokines is key to maintaining stem-like T cells in tumors.

It is often assumed that IFNγ directly inhibits cytotoxic T cells (CTL) cytokinesis or kill CTLs. However, CTLs and here, exhausted T cells, become less responsive to IFNγ as they down-regulate IFNγR2

**Fig. 5 | IFNγ sensing by CD8 T cells restricts proliferation by targeting stem-like T cells. a–c** Analysis of transcriptomics from Fig. 3. **a-** Percentages of CD8 T cells from Control (red) and CD8-IFNγRKO (grey) mice in G1, G2/M, or G1 phase of the cell cycle (*n* = 15706). (**b–c**) Normalized Enrichment score (NES) of cell cycle-related Hallmark (top) and Reactome (bottom) pathways in Stem-like (**b**) or Exhausted (**c**) T cell states. Bars are colored by adjusted p-values, derived from the fgsea R package (**d–f**) CFSE-labelled total immune cells from tumor-bearing WT mice were restimulated ex vivo with anti-CD3 and anti-CD28 and treated with blocking IFNγ antibody, recombinant IFNγ, or media only (Mock). Data are from 3 independent experiments. Each dot is one well of immune cells pooled from multiple mice. **d-** Ratio between Tcf1/7+ and Tcf1/7- TIL ratio over time (*n* = 7 for Day 0; *n* = 12 for Day 4.5 mock, *n* = 9 for Day 4.5 rIFNγ and Day 4.5 anti-IFNγ; *n* = 8 for Day 7 mock, *n* = 7 for Day 7 rIFNγ and Day 4.5 anti-IFNγ). Data shows Mean +/− SEM. Mixed-effects model analysis with Tukey's multiple comparison test. **e-** Absolute number of Tcf1/7⁺ PD-1⁺ stem-like T cells over time normalized to day 0 (*n* = 9 for Day 4.5 mock, *n* = 7 for Day 4.5 rIFNγ and Day 4.5 anti-IFNγ; *n* = 8 for Day 7 mock, *n* = 7 for Day 7 conditions). Data shows Mean +/− SEM. Mixed-effects model analysis with Šidák's multiple comparison test. **f-** Percentage of stem-like T cells that divided, assessed by CFSE dilution (*n* = 10 for Day 4.5 mock, *n* = 9 for Day 4.5 rIFNγ and Day 4.5 anti-IFNγ; *n* = 8 for Day 7 mock, *n* = 7 for Day 7 conditions). Data shows Mean +/− SEM. Mixed-effects model analysis with Šidák's multiple comparison test.

**g–i** Stem-like TILs (orange) and exhausted TILs (violet) from B16-OVA tumors bearing WT mice were labeled with CellTrace violet dye (CTV) and restimulated with anti-CD3 and anti-CD28. **g-** Representative examples of CTV dilution after 4 days. Control represents TILs at day 0. **h-** Percentage of dividing cells. Each point represents one individual sample (*n* = 8 for Stem-like; *n* = 10 for exhausted), from 2 independent experiments. Data shows Mean +/− SEM. Unpaired *t* test. **i-** Percentage of cells that completely diluted CTV. Each point represents one individual sample (*n* = 7), from 3 independent experiments. Data shows Mean +/− SEM. Unpaired *t* test. **j–k** CTV-labeled exhausted TILs from B16-OVA tumors were restimulated as in (**g–i**) in the presence (grey) or absence (red) of anti-IFNγ. **j-** Representative examples of CTV dilution profiles. **k-** Percentage of exhausted T cells that divided after 4 days. Each point represents one individual sample (*n* = 7), from 2 independent experiments. Data shows Mean +/− SEM. Unpaired *t* test. **l–n** In vitro-generated stem-like and exhausted IFNγRKO and WT OTI cells were ad-mixed and injected into B16-OVA tumor-bearing mice. **l-** Experimental set-up, created with BioRender.com. **m-** Expression of selected exhaustion markers and Tcf1/7 of in vitro differentiated WT and IFNγRKO OTI before transfer. Mock represents unstained OTI T cells. **n-** Ratio between IFNγRKO and WT OTI from stem-like (orange) or exhausted (violet) conditions in tumors 6 days after transfer. Each point represents one individual mouse (*n* = 20 for Stem-like; *n* = 12 for exhausted), from 3 independent experiments. Data shows Mean +/− SEM. Non-parametric Mann-Whitney test.

(Figs. 1A, B, 4D), critical for IFNγ-downstream signalling. This has also been shown in the context of infection and in human CD8 T cells[36,74]. We discovered that IFNγ targets exhausted precursors, stem-like T cells, which quickly differentiate into intermediate exhausted state upon restimulation, explaining how IFNγ indirectly affects exhausted T cells. Stemness and high proliferative capacity following restimulation, although counter-intuitive, are both hallmarks of stem-like T cells[46,75]. Our data suggest that they are both limited by IFNγ. This is consistent with the role of IFNγ in regulating the stemness of other cell types. IFNγ has a complex impact on hematopoiesis during inflammation[76], where, for instance, it impairs the maintenance of hematopoietic stem cells by directly inhibiting their proliferation and restoration upon viral infection[77]. In cancer stem cells, high doses of IFNγ inhibit self-renewal and induce apoptosis of colon cancer cell lines[78], whereas low doses of IFNγ potentiate the stemness of non-small cell lung cancer-derived cell lines, as well as dormancy in melanoma and breast cancer[79,80]. This highlights the important and likely dynamic crosstalk between tumor progression and the state of the immune response[81]. It is interesting to note that the level of IFNγR expression in cancer cells has been hypothesized to mediate either apoptosis or entry into a quiescent state[81]. Similarly, we found that the level of expression and ratio between IFNγR1 and IFNγR2 is tightly regulated and differs between T cell subsets (Figs. 1 and 4). As such, the dynamic regulation of IFNγR might lead to different IFNγ sensitivity and IFNγ-driven outcome.

Our data highlight the dual pro- and anti-tumor functions IFNγ exerts on T cells. On one hand, it potentiates T cell homing to tumors. On the other hand, IFNγ sensing by CD8 T cells results in their exclusion from the tumor core, which might occur by the same mechanism, whereby CD8 T cells produce IFNγ at the margin, where they are primed by macrophages, which in turn would induce the expression of the chemokines CXCL9 and 11[82], recruiting and retaining CD8 T cells that express the receptor CXCR3. Interestingly, the effect of IFNγ in increasing homing is counter-balanced by the effect IFNγ in inhibiting stem-like T cell maintenance and subsequent proliferation throughout their progeny. Whether the niches where stem-like T cells reside can to some extent shield stem-like T cells from IFNγ is unknown, but suggests that characterizing this niche will be key to decipher additional factors controlling stem-like T cell maintenance.

Partial agonists for the IFNγR have shown the capacity to uncouple the immunostimulatory from the immunosuppressive function of IFNγ[83], suggesting that their downstream pathways are distinct and therefore potentially targetable. Our work puts IFNγ at the center of

continuous T cell dysfunction and reinforces the importance to uncouple different outcomes triggered by IFNγ to specifically leverage IFNγ-driven anti-tumor immunity.

## Methods
### Mice
Mice were bred and maintained in the University of Oxford specific pathogen-free (SPF) animal facilities. Mice were routinely screened for the absence of pathogens and were kept in individually ventilated cages with environmental enrichment at 20–24 °C, 45–65% humidity with a 12 h light/dark cycle (7am–7pm) with half an hour dawn and dusk period. CD8α-Cre mice (The Jackson Laboratory – 008766) were crossed with Rosa-floxSTOPflox-Tomato (The Jackson Laboratory – 007908) and flox-IFNγR1-flox (The Jackson Laboratory – 0025394) to specifically deplete IFNγR1 in mature CD8 + T cells. IFNγR-/- (The Jackson Laboratory – 003288) were crossed with OTI mice (The Jackson Laboratory – 003831) to generate IFNγR-/- OTI. They were crossed with Ubiquitin-GFP mice (The Jackson Laboratory – 004353) or CD45.1 mice (The Jackson Laboratory – 002014). 6 to 10-weeks old WT recipient C57Bl6 mice were purchased from Charles River (027). Tcf1/7 GFP reporter mice (The Jackson Laboratory – 030909) were used as hosts for some experiments. Mice were housed and bred under specific pathogen-free/SPF conditions in the in-house animal facilities at the University of Oxford. Experimental and control animals were co-housed. Mice were euthanized by CO₂ asphyxiation followed by cervical dissociation. 6 to 10-weeks old male and female mice were used for all experiments. All experiments involving mice were conducted in agreement with the United Kingdom Animal Scientific Procedures Act of 1986 and performed in accordance with approved experimental procedures by the Home Office and the Local Ethics Reviews Committee (University of Oxford) under UK project license P4BEAEBB5 and PP3609558.

### Tumor induction
B16 Tyr-/- expressing mCherry and Ovalbumin (B16-OVA) was kindly given by Dr. Edward Roberts from the Beatson Institute (Glasgow, UK). Cells were cultured at 37C in 5% CO2 in Dulbecco's Modified Eagle's Medium (Sigma, D6429) media containing 10% fetal bovine serum (FBS) (Sigma, F9665), 0.5 μM beta-mercaptoethanol (Gibco, 31350-10), and Penicillin/Streptomycin (Gibco, 10378-016). Between $2 \times 10^5$ to $5 \times 10^5$ B16-OVA or MC-38 cells were resuspended in phosphate-buffered saline (PBS) and 25% Matrigel (Corning, 354262) for subcutaneous injection in the flank of host mice. Tumors were palpable after 5 to 7 days post-engraftment and were measured using a caliper

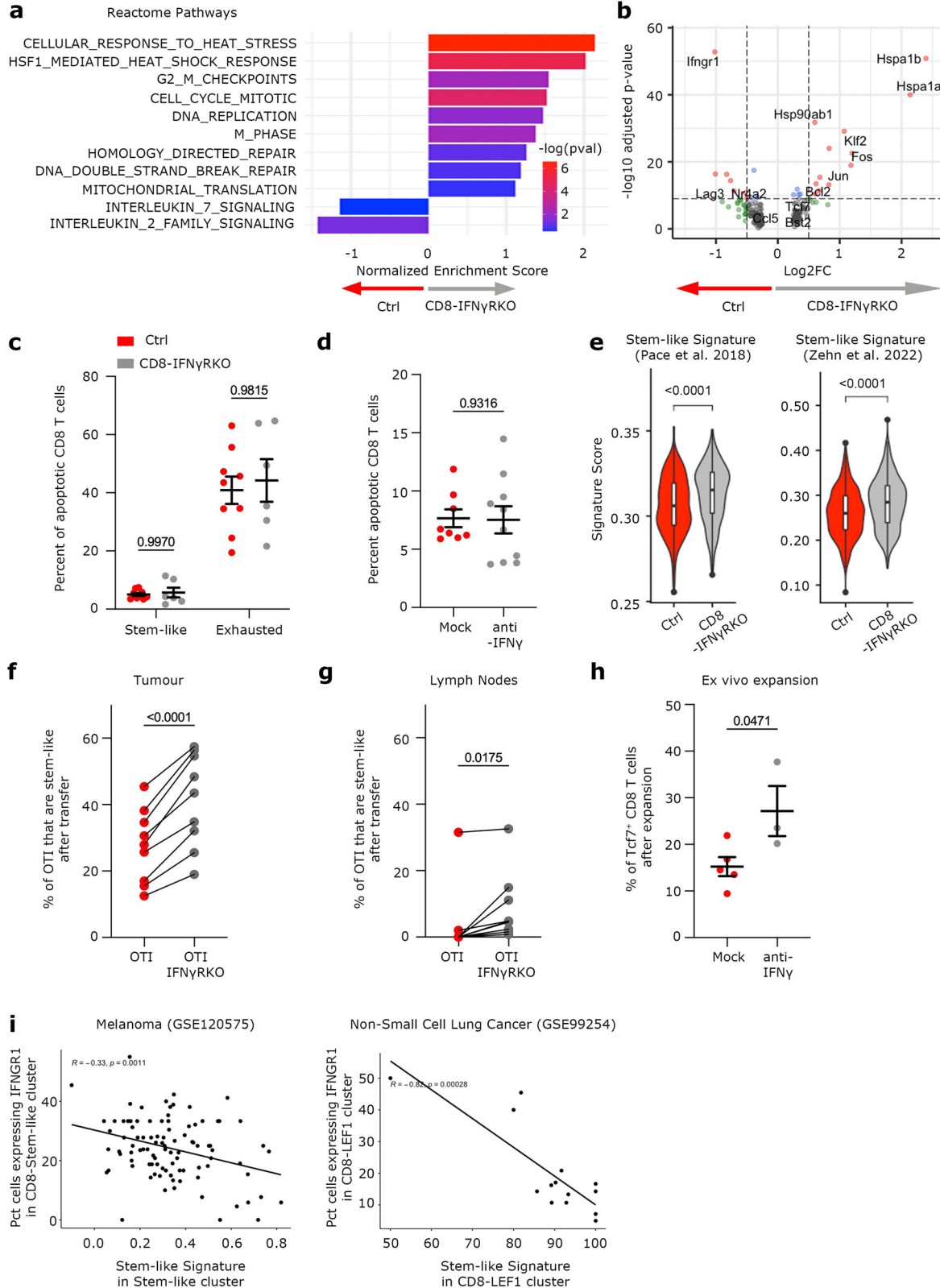

and the [(Length*Width*Width)/2] formula was used to obtain the tumor volume in mm³.

### Adoptive T cell transfer of naïve CD8 T cells

WT or IFNγRKO OTI cells were isolated from the lymph nodes of 6 to 12 weeks-old mice. CD8 T cells were negatively selected using a negative mouse CD8 T cell isolation kit (Mojosort – Biolegend). When indicated, cells were labeled with 5 µM CellTrace Violet (Thermo Fisher) for 15 min at 37 °C. Mice were adoptively transferred with 2×10⁶ OTI by intravenous injections between 5 and 7 days after B16-OVA cell engraftment when the tumor was palpable depending on the experiment.

**Fig. 6 | IFNγ signaling in intratumoral Stem-like CD8 T cells impairs their maintenance. a–b; e** Analysis of transcriptomics data from Fig. 3. **a**- NES of chosen Reactome pathways up- and down-regulated in CD8-IFNγRKO vs control Stem-like T cells. Bars are colored by adjusted p-values, derived from the fgsea R package. **b**- Volcano plot of Differentially Expressed Genes between control and CD8-IFNγRKO Stem-like T cells. Green dots: genes with log2 (fold-change) value >0.5 or <−05; Blue dots: genes with an adjusted *p* value < 0.05; Red dots: genes with log2 (fold-change) value >0.5 or <−05 and an adjusted *p* value < 0.05. Total variables = 195. Wilcoxon rank sum test. **c**- Control (red) and CD8-IFNγRKO (grey) mice were engrafted with B16-OVA tumors. Percentage of apoptotic cleaved Caspase 3[+] stem-like or exhausted TILs. Each dot represents an independent mouse (*n* = 9 for Control, *n* = 6 for CD8-IFNγRKO), from 2 independent experiments. Data shows Mean +/− SEM. Mixed-effects model analysis with Šidák's multiple comparison test. **d**- Naïve CD8 T cells were stimulated with anti-CD3, anti-CD28 and IL-2 and treated with anti-IFNγ when indicated. Percentage of cleaved-Caspase 3[+] apoptotic CD8 T cells. Each dot represents a mouse (*n* = 8 for Mock and *n* = 10 for anti-IFNγ) from 2 independent

experiments. Data shows Mean +/− SEM. Unpaired *t* test. **e**- Stem-like signature scoring of Control (red) and CD8-IFNγRKO (grey) stem-like cells. Box plots indicate median (middle line), 25th, 75th percentile (box). Pairwise group comparisons with two-sided Wilcoxon signed-rank test (*n* = 15706 total). **f–g** In vitro-generated stem-like WT (red) and IFNγRKO (grey) OTI cells were injected at a 1:1 ratio in tumor-bearing mice as described in Fig. 5L. Percentage of PD-1[+] Tcf7[+] OTI in tumors (**f**) and in draining lymph nodes (**g**) after 6 days. Lines link cells from the same mouse. Each dot represents a mouse (*n* = 9) from 3 independent experiments. Data shows Mean +/− SEM. Paired *t* test. **h**- stem-like TILs were sorted and restimulated with anti-CD3 and anti-CD28 and treated with anti-IFNγ when indicated. Percentage of PD-1[+] Tcf7[+] cells after 4 days. Each dot is a mix of 5 mice (*n* = 5 for Control; *n* = 3 for CD8-IFNγRKO) from 3 independent experiments. Data shows Mean +/− SEM. Paired *t* test. **i**- Correlation between the percentage of IFNγR1 expressing stem-like T cells and Stem-like signature scoring in transcriptomics data from human tumor patients. R = Pearson's correlation coefficient; Fisher's test. Each dot is a patient (left *n* = ; right *n* = 14).

## Adoptive T cell transfer – Tracing

For cell-tracing experiments, OTI cells from both WT or IFNγRKO were in vitro-treated to induce stem-like T cells or exhausted T cells. To obtain stem-like T cells, $2 \times 10^6$ OTI/mL were incubated 24 h at 37 °C in RPMI 1640 media (Gibco) containing 10% fetal bovine serum (FBS) (Sigma), 0.5 μM beta-mercaptoethanol (Gibco) and Penicillin/Strep-tomycin (Gibco) (called R10 thereafter) supplemented with 10IU/ml of IL-2, 100 ng/mL of N4 peptide (SIINFEKL) and 7uM of TWS119 (Merk)[48]. To obtain in vitro induced exhausted T cells, 500 000 OTI/mL were plated in R10 containing 1% HEPES (Gibco), 5 ng/mL IL-15 (Peprotech), 5 ng/mL IL-7 (Peprotech), and 10 ng/mL of N4 peptide (SIINFEKL). Cells were incubated at 37 C and 10 ng/mL of N4 peptide (SIINFEKL) were added daily for 5 days. Cells were split with fresh media when confluent and underwent Ficoll separation (Ficoll-paque Premium, Cytiva) prior to in vivo injection[47]. Mice were adoptively transferred with $2 \times 10^6$ OTI WT and IFNγRKO at 1:1 ratio by intravenous injections 6 days after B16-OVA cell engraftment.

## Adoptive T cell transfer – Homing

Naïve WT OTI and IFNγRKO OTI resuspended in R10 supplemented with 2 ug/mL of anti-CD28 (PV-1, Biolegend) were plated on a previously coated plate using a 5 ug/mL solution of anti-CD3 (2C11, Biolegend) and incubated at 37 °C for 7 days. After 2 days of culture, 5IU/mL of IL-2 was added with fresh media. After 7 days, activated cells were labeled with CellTrace Violet before mixing the OTI WT and IFNγRKO at a 1:1 ratio. A total of $30 \times 10^6$ cells ($15 \times 10^6$ of each cell type) were injected intravenously in PBS on tumor-bearing mice 10 days after engraftment. Tissues were harvested after 24 h.

## Tissue processing

At the indicated time points, tumors and draining lymph nodes (LNs) were harvested, measured, and weighed. Tumors were then dila-cerated, incubated 30 min at 37 °C in 5% CO2 in R10 containing Lib-erase TL at 1 mg/mL (Roche) and DNase I at 10 μg/mL (Roche), and run through a 70μm cell strainer (Falcon). Cells were either undergoing Ficoll to obtain live immune cells or directly used after for further experiments. LNs were directly smashed through a 70μm cell strainer and resuspended in PBS.

## Ex vivo T cell re-stimulation for cytokine detection

For OTI cells, Tumors from OTI-bearing mice were harvested 5 to 7 days after OTI transfer and processed as above. For endogenous TILs, tumors from control or CD8-IFNγRKO mice were harvested 10 to 12 days after tumor engraftment. Single-cell suspensions were ficolled and plated in 96-well plates. Cells were incubated for 6 h at 37 °C in the presence or absence of the OVA peptide SIINFEKL (1 ug/mL). Brefeldin A (BFA, 10 ug/mL, Santa Cruz) was added after 30 min.

## Ex vivo T cell self-renewal assessment

TILs were isolated from tumor-bearing WT mice, enriched using Ficoll, and labeled with 2uM of CFSE. Between $2 \times 10^5$ to $2 \times 10^6$ cells were resuspended in complete RPMI supplemented with 2 ug/mL of anti-CD28 (Biolegend-37.51) and with or without blocking IFNγ antibody (BioXcell XMG1.2 at 10 ug/mL) or recombinant mouse IFNγ (Biolegend at 100 ng/mL). Cells were then plated at a similar density in 96-well or 12-well plates previously coated with 2 ug/mL of anti-CD3 (Biolegend – 145-2C11) in PBS for 2 h at 37 °C. Cells were incubated at 37 °C and were fed every 2 days with fresh media according to the different conditions. At the desired time points cells were harvested and processed for flow cytometry.

In some experiments, naïve T cells were stimulated by 2 μg/ml coated anti-CD3 and 2 μg/ml soluble anti-CD28 with or without blocking IFNγ antibody as above and treated with 3.5 μM of the inhibitor TWS119 (Merk) when indicated. IL-2 at 10IU/ml was added to the culture two days after priming.

## Ex vivo T cell proliferation assessment

TILs from tumor-bearing Tcf1/7-GFP mice were enriched using a CD45 (TIL) Microbeads enrichment kit (Miltenyi Biotech) and stained for cell sorting. Stem-like T cells and exhausted T cells were then sorted from live CD8 + PD1 + T cells using GFP + LAG3- strategy for stem-like T cells and GFP-LAG3+ for exhausted TILs. Cells were labeled with 5 μM CellTrace Violet (Thermo Fisher) for 15 min at 37 °C. Stem-like T cells and exhausted T cells were resuspended separately in R10 supplemented with 2 ug/ml soluble anti-CD28 with or without blocking IFNγ antibody at 10 ug/mL and plated in a 96-well plate pre-viously coated with 2 μg/ml of anti-CD3. Cells were processed for flow cytometry after 4 days of culture to assess their proliferation.

## WNT pathway analysis in vitro

CD8 T cells were isolated from OTI mice as described above. Depending on the experiment, naïve cells were pre-treated with 100 ng/mL of recombinant IFNγ or not for 16 h at 37 °C, before being incubated at 37 °C for 6 h in R10 only or R10 supplemented with 3.5 μM of TWS119 or 400 ng/mL of WNT3a (Peprotech). During this incuba-tion, some cells were plated on anti-CD3 coated wells and supple-mented with 2 μg/ml of anti-CD28 for TCR activation.

## In vivo cytotoxicity assessment

WT or IFNγRKO OTI cells were transferred in tumor-bearing WT mice between 5 and 7 days after B16-OVA cell engraftment when the tumor was palpable. After 7 days of OTI adoptive transfer, mice were injected with a 1:1 ratio of Ova-targetable and non-targetable cells. Splenocytes from naïve B6 mice were isolated and split into two equal pools. Tar-getable cells were loaded with SIINFEKL peptide (1 μg/mL) and further labeled with high concentration CFSE (2 μM - Invitrogen). Non-

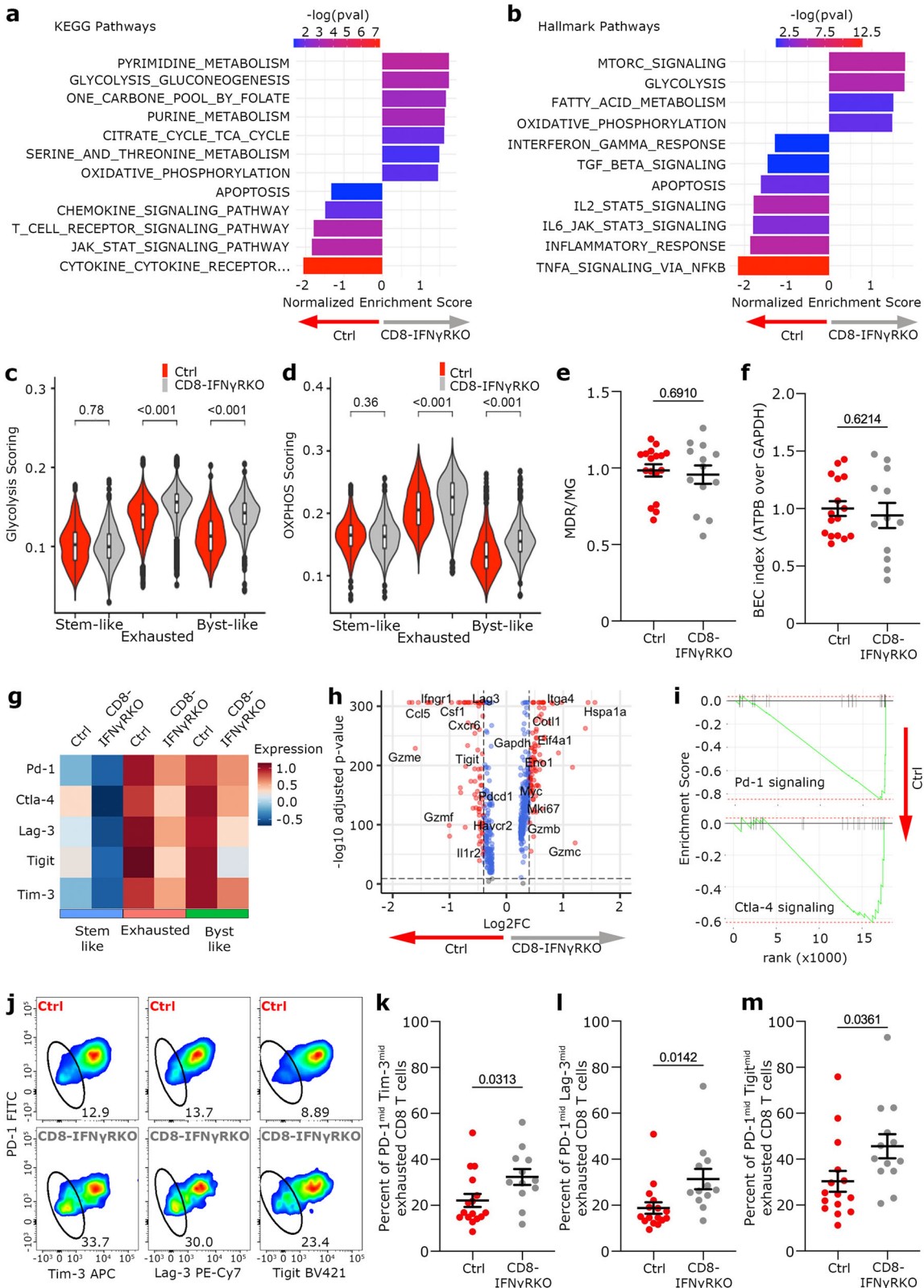

targetable cells were not loaded but labeled with low CFSE (0.2uM). Peptide loading was performed in complete RPMI at 37 °C for 1 h. Cell labeling was done in PBS for 15 min at 37 °C. After cell preparation, cells were washed with PBS and pooled together for intravenous injections. After 24 h spleens were harvested and processed for flow cytometry analysis.

**Tumor processing for confocal imaging**

Tumors were harvested from day 8 to day 20 after engraftment and immersed in a commercial paraformaldehyde-based fixative solution (Antigenfix – Diapath P0016) overnight under gentle agitation at 4 °C protected from light. Tumors were then rinsed with PBS and incubated in PBS 30% sucrose (VWR chemicals) for a minimum of 6 h at 4 °C

**Fig. 7 | IFNγ signaling in TILs is not sufficient to alter their metabolism but reinforces exhaustion in vivo. a–d; g–i** Analysis of transcriptomics data from Fig. 3. **a–b** NES of chosen KEGG (**a**) and Hallmark (**b**) pathways differentially regulated in CD8-IFNγRKO vs control Exhausted TILs. Bars are colored by adjusted *p*-values, derived from the fgsea R package. **c–d** Violin plots show Glycolysis scoring (**c**) and Oxphos scoring (**d**) of Control (red) and CD8-IFNγRKO (grey) TILs within each cell state. Box plots indicate median (middle line), 25th, 75th percentile (box). Pairwise group comparisons with two-sided Wilcoxon signed-rank test. (*n* = 15706 total). **e–f** Metabolic analysis of PD-1⁺ Lag-3⁺ TILs from Control and CD8-IFNγRKO mice engrafted with B16-OVA. Each dot represents a mouse (Control *n* = 17 and CD8-IFNγRKO *n* = 13) from 3 independent experiments. Data shows Mean +/− SEM. Unpaired *t* test. **e**- Ratio between MitoTracker Deep red (MDR) and MitoTracker green (MDG). **f**- BEC index (ratio between ATPB and GADPH expression). **g**- Heatmap shows the relative average expression of selected exhaustion markers in control and CD8-IFNγRKO TILs over the different cell states. **h**- Volcano plot

representing Differentially Expressed Genes between control and CD8-IFNγRKO TILs in Exhausted T state. Blue dots: genes with an adjusted p value<0.05; Red dots: genes with log2 (fold-change) value >0.5 or < −05 and an adjusted p value<0.05. Total variables = 543. Wilcoxon rank sum test. **i**- Enrichment Score of PD-1 (upper panel) and CTLA-4 (lower panel) between CD8-IFNγRKO and control TILs in the Exhausted state. **j–m** Phenotypic analysis of PD-1⁺ Lag-3⁺ TILs from Control and CD8-IFNγRKO mice engrafted with B16-OVA. Each dot represents a mouse from 2 independent experiments. Data shows Mean +/− SEM. Unpaired *t* test. **j**- Representative flow cytometry plots with gate highlighting a population with intermediate expression (mid) of PD-1, Tim-3, Lag-3, and Tigit. **k–m** Percentage of TILs with intermediate expression of PD-1 and Tim-3 (**k**; Control *n* = 16 and CD8-IFNγRKO *n* = 12), Lag-3 (**l**; Control n = 16 and CD8-IFNγRKO *n* = 12) and Tigit (**m**; Control *n* = 15 and CD8-IFNγRKO *n* = 13). Each dot represents a mouse from 3 independent experiments. Data shows Mean +/− SEM. Unpaired *t* test.

without agitation. Fixed tumors were then embedded in O.C.T. compound (VWR chemicals) and kept frozen at −80 °C. Frozen tumors were sectioned with a cryostat (Leica - CM1900UV) to obtain 10µm cryosections. Cryosections were rehydrated in PBS for 5 min and blocked with PBS 10% donkey serum (Sigma), Fc-block (anti-CD16/32 clone 93 Biolegend) 2% FBS, and 0.1% triton (Acros Organics) for 4 h at room temperature. The sections were then incubated with anti-CD8α-A647 at 1/200 diluted in the blocking solution (Abcam – ab237365) overnight at 4 °C, washed, and stained with the secondary antibody anti-rabbit-A647 at 1/400 (Biolegend - polyclonal) for 4 h at room temperature. Sections were washed and incubated in DAPI (Sigma) at 0.5 µg/ml for 15 min at room temperature and were mounted with Fluoromount G (SouthernBiotech). Images were taken on the Zeiss LSM880 confocal microscope and analyzed with the Imaris (Bitplane v9.6) software.

## Flow cytometry

Cells were washed with PBS and incubated for 30 min at 4 °C with live-dead stain (Zombie-NIR Biolegend, Fixable Viability Dye eFluor® 780 Invitrogen, or Live/dead-Aqua Invitrogen) at 1/1500 with Fc-block at 1/200. For metabolic dyes, cells were incubated in complete R10 at 37 °C with either NAO (Nonyl Acridine Orange – Invitrogen) for 10 min or with MitoTrackers (Deep Red – Invitrogen, and Green – Invitrogen) for 30 min before the extracellular staining. Extracellular staining was then performed with directly conjugated antibodies from Biolegend against CD45 (30-F11), CD8 (53–6.7), CD44 (IM7), CD69 (H1.2F3), F4/80 (BM8), CD3 (1782), NK1.1 (PK136), CD19 (6D5), CD4 (RM4-5), CD11b (M1/70), CD11c (N418), Ly6C (HK1.4), Ly6G (1A8), MHC-II (M5/114.15.2), CD45.1 (A20), CD45.2 (104), LAMP1 (1D4B), LAG3 (C9B7W), PD1 (29 F.1A12), TIM3 (RMT3-23), TIGIT (1G9) for 30 min at 4 °C. Cells were then fixed with PBS 4% Paraformaldehyde (Thermo Fisher Scientific) for 15 min at room temperature. Note, for the metabolic dyes, cells were kept alive and not fixed before running them by flow cytometry. For intracellular cytokine and metabolic markers staining, after extracellular staining, cells were fixed and permeabilized with BD Cytofix/Cytoperm kit (BD Biosciences) and incubated with fluorescently conjugated antibodies against IFNγ (XMG1.2) and TNFα (MP6-XT22) for 45 min at RT for the cytokine panel. And incubated with fluorescently conjugated antibodies against GLUT1 (EPR3915), ATPB (3D5), GADPH (14C10), LDHA (E9) for 20 min at room temperature for the metabolic panel. For Tcf1/7, cleaved caspase 3, and β-catenin staining, cells were fixed with PBS 4% Paraformaldehyde (Thermo Fisher Scientific), permeabilized with cold methanol (Fisher Chemical) for 10 min and incubated with fluorescently conjugated Tcf1/7 antibody (clone C63D9, Cell Signalling), pan β-catenin (clone 15B8, Invitrogen) or cleaved caspase 3 (clone C92-605, BD Bioscience) for 1 h at room temperature. Samples were recorded on a BD-LSR Fortessa X-20 and recorded using BDFACSDiva (v8.0) software. Data analysis was performed using FlowJo v.10.4.2 (FlowJo LLC). All antibodies dilution can be found in Supplementary Dataset 4.

## Cell sorting for single-cell sequencing and library preparation

All cells from B16-OVA tumors grown 3 controls and 3 CD8-IFNγRKO mice were stained with extracellular markers as described above, without fixation step. Cells were then sorted based on the expression of CD45, CD3, CD8, and Tomato using a FACSAria™ II (BD).

Approximately 20,000 cells per sample were loaded onto the 10X Genomics Chromium Controller (Chip K). Gene expression, feature barcoding and TCR sequencing libraries were prepared using the 10x Genomics Single Cell 5' Reagent Kits v2 (Dual Index) following manufacturer user guide (CG000330 Rev B). The final libraries were diluted to ~10 nM for storage. The 10 nM library was denatured and further diluted prior to loading on the NovaSeq6000 sequencing platform (Illumina, v1.5 chemistry, 28 bp/98 bp paired end for gene expression and feature barcoding, 150 bp paired end for TCR libraries).

## Mouse scRNAseq and analysis

Sequence reads were mapped using CellRanger multi (version 6.0.0) with the 10x mouse reference transcriptome (version 2020-A). Datasets were analyzed using Seurat version 4.0.6[84]. Samples were merged on a single Seurat object. We filtered out cells having less than 200 and more than 6000 detected genes, cells in which mitochondrial protein-coding genes represented more than 3% of UMI and that had more than 20% of large gene content. Cells were then further filtered based on the expression of *Cd2*, *Cd8a* and *Cd8b1*. Samples were normalized with the standard workflow (NormalizeData, FindVariableFeature, ScaleData), and variation associated with mitochondrial and ribosomal UMI percentage and cell cycle were regressed out. Principal components were calculated using the top 3000 variable features. These genes were used as input for principal component analysis (PCA), and significant PCs (*n* = 15) identified using Seurat ("JackStraw" test and "Elbowplot"). Samples were then integrated with Harmony[85]. Clustering was performed with the Louvain algorithm (*n* = 15 PCs, Resolution = 0.4). Significant differentially expressed genes between clusters were identified using the "FindAllMarkers" function, Wilcoxon test and selecting markers expressed in at least 25% of cells. This step revealed macrophage contamination in 1 cluster, which was filtered out and data was re-processed. The UMAP projection was computed using significant PCs ("RunUMAP" function, Seurat). Significant differentially expressed genes between genotype within clusters were identified using the "FindMarkers" function, Wilcoxon test and selecting markers expressed in at least 25% of cells. Pathway analysis was performed with Fast gene set enrichment analysis (fgsea), using the Gene Ontology or the Reactome pathway repositories. Gene signatures (Supplementary Dataset 5) were computed manually and analyzed using the packages Vision[86] and UCell which calculates module enrichment scores using Mann-Whitney U statistic[87]. The Cell cycle score was calculated using the Seurat function "CellCycleScoring". Cell Cycle score was assigned for each cell

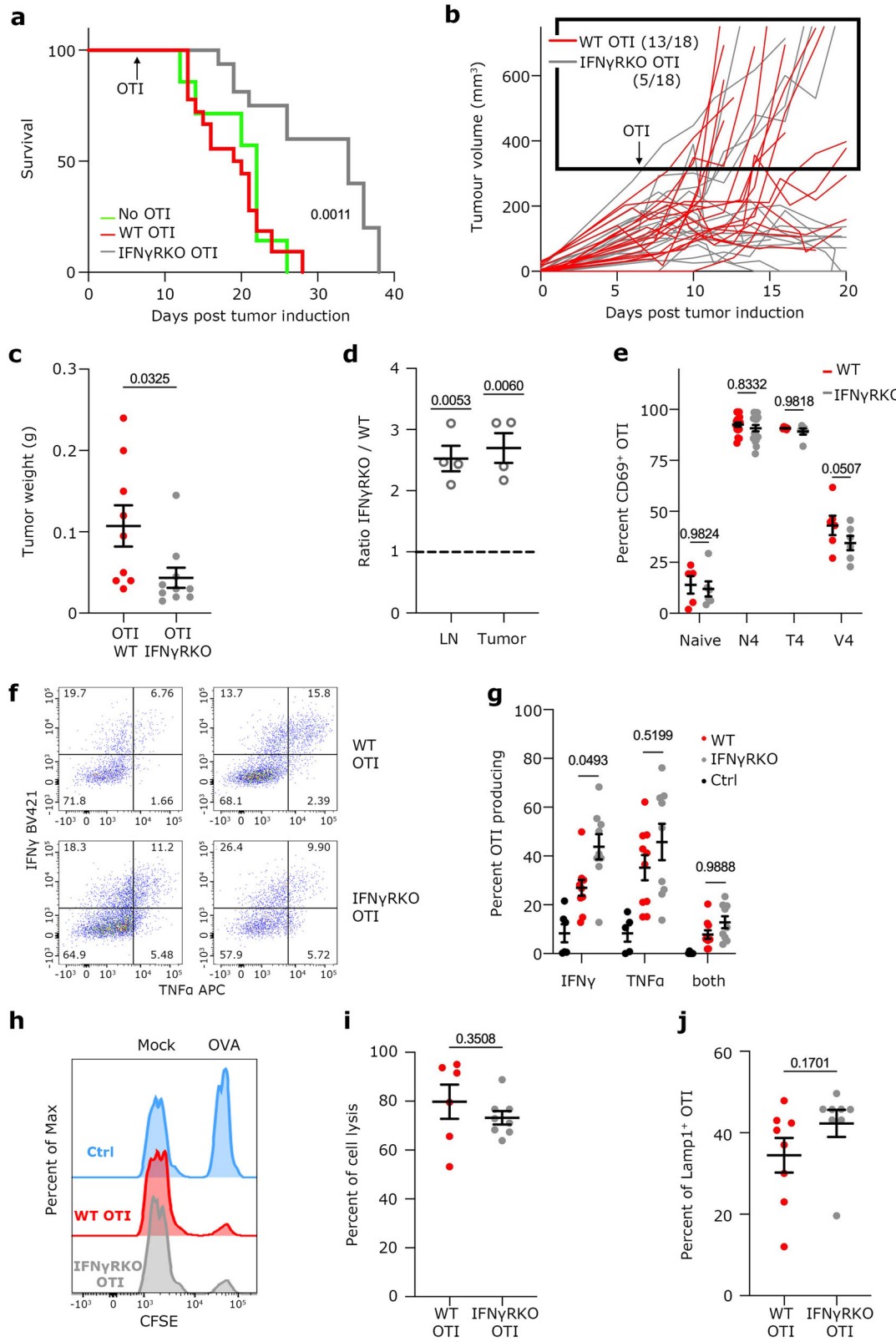

based on the expression of G2/M and S phase markers. Cells expressing neither were labelled as in G1 phase. Single-cell trajectory analysis was inferred using the Monocle 3 software[88]. Pseudotime inference was performed on the UMAP created previously in Seurat. The node within the stem-like cluster in the trajectory was selected as the root node. Data was visualized using EnhancedVolcano (v1.12.0), GGplot2(v3.3.5) and ggpubr(v0.5.0).

**Mouse TCR sequencing analysis**

Paired chain TCR sequences were obtained through targeted amplification of full-length V(D)J segments during library preparation. Sequence assembly and clonotype calling was carried out using the CellRanger immune profiling pipeline (cellranger multi). TCR profiling on filtered contig annotations was done using R package scRepertoire version 1.1.4[89]. Only cells for which both TCRα and

**Fig. 8 | IFNγR ablation in CD8 T cells improves tumor control in a T cell transfer model. a–c** WT mice bearing B16-OVA were transferred with Control (red), IFNγRKO (grey) OTI T cells or not transferred (green). **a**- Mouse survival over time ($n = 7$ control, n = 17 WT and n = 18 IFNγRKO). Data is from 3 independent experiments. Mantel-Cox test. **b**- Tumor volume of individual mice over time ($n = 15$). Black square highlight tumors with a volume of at least 300 mm³. Data is from 3 independent experiments. **c**- Average tumor weight 7 days post-OTI transfer. Each dot is a mouse ($n = 9$ WT and $n = 10$ IFNγRKO) from 2 independent experiments. Data shows Mean +/− SEM. Unpaired *t* test. **d**- WT mice bearing B16-OVA were either co-transferred with Control and IFNγRKO OTI T cells. Ratio between the number of IFNγRKO and WT OTI cells in Lymph nodes and Tumors after 5 days. Each dot is a mouse ($n = 4$) from 2 independent experiments. Data shows Mean +/− SEM. One sample *t* and Wilcoxon test. **e**- WT and IFNγRKO OTI were stimulated in vitro with BMDCs loaded with N4, T4, or V4 peptides. Percentage of CD69⁺ OTI cells after 24 h. Each dot is an independent sample ($n = 5$ WT-naive, $n = 6$ WT-T4/V4, IFNγRKO-naïve/T4/V4; $n = 18$ WT-N4 IFNγRKO-N4) from 2 experiments. Data shows Mean +/−

SEM. Mixed-effects model analysis with Šidák's multiple comparison test. **f–g** WT mice bearing B16-OVA tumors were transferred with Control (red) and IFNγRKO (grey) OTI T cells. Intra-tumoral OTI cells were restimulated in vitro with 1ug OVA peptide. **f**- Representative plot showing IFNγ and TNFα production. **g**- Quantification of IFNγ and TNFα production by OTI WT (red), OTI IFNγRKO (grey), and non-stimulated OTI control (black). Each dot is an independent sample ($n = 9$ WT, $n = 6$ IFNγRKO) from 3 independent experiments. Data shows Mean +/− SEM. Mixed-effects model analysis with Šidák's multiple comparison test. **h–j** WT mice bearing B16-OVA tumors were transferred with Control (red) and IFNγRKO (grey) OTI T cells. **h–i** In vivo cytotoxic assay was performed after 12 days. **h**- Representative histogram of CFSE labelled target cells after 24 h. **i**- Quantification of cell lysis. Each dot is an independent sample ($n = 6$ WT, $n = 8$ IFNγRKO) from 2 independent experiments. Data shows Mean +/− SEM. Unpaired *t* test. **j**- Lamp-1 expression following restimulation. Each dot is an independent sample ($n = 8$ WT, $n = 8$ IFNγRKO) from 2 independent experiments. Data shows Mean +/− SEM. Unpaired *t* test.

TCRβ could be identified were used. Clone calling was done for each sample set independently before integration in the Seurat object.

## Human dataset analysis

Single-cell RNAseq data from Watson et al (EGAS00001005507)[10] was analyzed for *IFNGR1/2* expression across labelled CD8 T cell subsets or clone size groupings. The cohort consisted of eight patients who were prescribed checkpoint blockade. Four participants received single-agent pembrolizumab and four received combination ipilimumab/nivolumab. Blood samples were collected immediately before treatment at d0 and d21 after 1 cycle of treatment[10]. CD8 T cell subset annotations were kept consistent with the annotated Seurat object from Watson et al, with the exception of MAIT cells, which were further annotated as cells carrying *TRAV1-2 TRAJ12/20/33* TCR chains within MAIT-containing Seurat clusters (1, 19, 20 and 24). Clone size groups were calculated as a proportion of the TCR repertoire per sample, in accordance with the original definitions. Cells with above-zero normalized expression of IFNGR1/2 were defined as *IFNGR1/2*-expressing.

Expression of *IFNGR1/2* in bulk RNAseq data from CD8 T cells was analyzed using data from Watson et al (EGAS00001004081)[10]. Samples consisted of blood CD8 T cells from individuals taken at 21 days after treatment ($n = 110$) for which clinical follow-up was available, as described in Watson et al[10]. Expression of *IFNGR1/2* was analyzed using DESeq2-normalized expression data. To identify pathways that correlated with IFNGR1/2 expression, normalized *IFNGR1/2* expression was used as a continuous variable in DESeq2 (v.1.38.1). Comparisons between individuals with divergent 6-month progression and auto-immune toxicity outcomes were performed using Wilcoxon rank sum tests.

Multiple datasets were analyzed to assess the association between *IFNGR1* and stem-like signature in human tumor patients. The melanoma dataset (GSE120575)[90] was filtered for CD8-expressing cells. Data was log-normalized, the cells were clustered, and the analysis focused on the *CCR7*-positive, *PDCD1*-low cluster. The level of expression of *IFNGR1* and stem-like signature scoring was computed within this cluster was analyzed for each patient. For the non-small cell lung cancer dataset (GSE99254)[91], cells from the "CD8_C1-LEF1" cluster were subsetted and the percentage of *IFNGR1* expressing cells and stem-like signature score were calculated for each patient.

## Statistical analysis

Data represent mean + standard error of the mean (sem) unless specified. Statistics were done with GraphPrism software unless specified. Comparisons between 2 groups were analyzed with t-test. Comparisons between more than groups were analyzed with one-way or two-way ANOVA. Survival data has been analyzed with Wilcoxon test. Growth curve over time were analyzed with two-way ANOVA. Analysis of the difference between the proportion of cells in clusters between two

scRNA-seq samples was analyzed with the r package "scProportionTest", which uses permutation test to calculate the p-value for each cluster/cell state, and a confidence interval for the magnitude difference. Data were considered significant when P values were 0.05 or less.

## Reporting summary

Further information on research design is available in the Nature Portfolio Reporting Summary linked to this article.

## Data availability

The mouse scRNAseq and scTCRseq data generated in this study have been deposited in the GEO database under accession code GSE221118. The following published datasets were reused in this study: EGAS00001005507, EGAS00001004081, GSE120575, GSE99254. All data are included in the Supplemental Information or available from the authors upon reasonable requests, as are unique reagents used in this Article. The raw numbers for charts and graphs are available in the Source Data file whenever possible. Source data are provided with this paper.

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

## Acknowledgements

We would like to thank Ed Roberts and Elizabeth Thompson for critical reading of the manuscript, the Wellcome Trust Centre for Human Genetics for the generation of the sequencing data, J. Webber for the assistance with cell sorting, and the dynamic platform and microscopy facility at the Kennedy Institute. This work was supported by Cancer Research UK (CR-UK) (C5255/A18085 through the Cancer Research UK Oxford Centre and 29549 to A.G); the Kennedy Trust for Rheumatology Research (KENN151607 and KENN202112 to A.G), the BBSRC (BB/R015651/1 to A.G), John Fell Funds (0006162 to A.G), and Kennedy Studentship (to J.M.M), Welcome Trust studentship (V.W.C.L).

## Author contributions

J.M.M and J.N.M performed all experiments, except as noted thereafter; B.P.F., R.A.W. and O.T. conducted computational re-analyses of the human dataset; M.A. generated mouse scRNA-seq libraries; L.K. performed the pre-processing steps to obtain the matrix files for mouse computational analyses. S.N.S supervised M.A. and L.K.; A.G. and V.W.C.L. performed subsequent mouse computational analyses; V.W.C.L. performed or assisted with some experiments. A.V.L-V. designed, analyzed and helped perform metabolism experiments. G.P. performed homing experiments. A.G. contributed to conceptualization,

funding acquisition, project administration, supervision, and writing. All authors edited the manuscript.

## Competing interests

The authors declare no competing interests.
