## [Peer Review File · Nature Communications]

We would like to thank the reviewers for their thorough assessment of our manuscript. We have now addressed their comments, and we believe the manuscript has consequently greatly improved. All changes are highlighted in the manuscript in yellow. Please find below our point-by-point response to the reviewers' comments:

Reviewer #1 (Remarks to the Author):

Interferon gamma is known to have both positive and negative regulatory impact on CD8 T cell immunity, in a context-dependent manner. The authors provided new evidence supporting the negative effect on anti-tumor immunity in treating melanoma. The biological data are convincing, in showing IFN γ R1 KO CD8 T cell expanded better, infiltrated the tumor with higher frequency, while cytotoxicity largely remained the same. The authors also observed the negative correlation in metastatic melanoma patients. These are solid data, although conceptually not entirely new, given the existing knowledge.

Thank you for your overall insightful comments on our manuscript. We now believe that our new data provide evidence that IFN γ sensing by stem-like T cells is the key mechanism restricting anti-tumor immunity and that our manuscript provides a mechanism by which stem-like T cells are regulated, which in itself is new. In addition, we now demonstrate that IFN γ inhibits not only proliferation, but also T cell diversity and stemness maintenance, which to our knowledge, is also novel.

The authors sought to understand the underlying mechanism, by performing single cell RNAseq on tumor infiltrating WT and IFN γ R1 KO CD8 T cells. The authors concluded that IFN γ inhibited the expansion of stem-cell subset, therefore compromising anti-tumor immunity. Based on the data provided, the author's interpretation may only represent one scenario, the supporting evidence for the conclusion on stem-like cells is not sufficiently strong, and alternative interpretation cannot be excluded.

We agree with the reviewer that other mechanisms are simultaneously at play. We believe that the advantage of the revised study is that we could dissect those inter-related mechanisms. We have now addressed the function of IFN γ on stem-like T cells in a multiple of experiments, from TCR sequencing, scRNAseq, phenotyping, functional experiments and cell tracing. We have also addressed the relevance of other mechanisms and concluded that IFN γ sensing by CD8 T cells mainly affects stem-like T cells, their maintenance and proliferation. Important to our conclusion, we have excluded the fact that IFN γ R deletion enhances homing. IFN γ RKO T cells actually have a strong defect in homing (Fig.S2F). We also have excluded the fact that IFN γ directly inhibits exhausted T cell proliferation. For example, anti-IFN γ treatment does not enhance exhausted T cell proliferation ex vivo (Fig.5J-K). We finally performed multiple experiments showing that stem-like T cell phenotype is retained in cells that cannot sense IFN γ in vivo, ex vivo and with tracing experiments (Fig.5D-F, 5L-N, 6C, 6F-H, S6F, S7I).

Please see below and the edited manuscript for detailed explanation.

Major issues:

1. Based on scRNAseq, the stem-like population represent only a minor population (<10%), and the difference of this subset between WT and KO is really small. The subset composition changes were never validated on tumor infiltrating CD8 cells using any of the models.

We have validated our data by flow cytometry (Fig.4C). In addition, we also provided evidence that in vitro treatment of TILs by anti-IFN γ also leads to an increased proportion of cells that maintained their stemness phenotype (Fig.5D-F). We also provided evidence using transcriptomics data from human cancer that IFN γ R expression anti-correlates with stem-like signature (Fig.6I and new Fig.1F and 6E). We now could also demonstrate that ex-vivo stem-like T cell maintenance is inhibited by IFN γ (Fig.6H). Finally, we can now show that in vitro-generated stem-like T cells have a greater expansion capacity if they do not sense IFN γ (Fig.5L-N), and increase stemness maintenance in vivo (Fig.6F-G) Altogether, we hope it convinces the reviewer that IFN γ indeed restricts stem-like T cell maintenance.

2. The authors showed the expansion capacity using in vitro stimulation model (Fig. 6c), although antiCD28 is included in the culture (sometimes antiCD3), this does not reproduce the microenvironment.

We apologise for the confusion, anti-CD3 and anti-CD28 were always added in combination in our cultures. We went through the manuscript to ascertain we did not forget to include this information in some places. When we initially assessed tumour-infiltrating T cells proliferation and stemness *ex vivo* (now appears in Fig.5D-F), we ficolled the cells to remove dead cells, debris and most cancer cells, obtaining a culture of mixed cells instead of a pure population to retain most of the tumor immune landscape. We made it clear in the revised manuscript: “To get further insight into relationship with cell states during proliferation, we isolated all tumor-infiltrating immune cells from WT mice, treated those with anti-CD3 and anti-CD28, and assessed how IFN γ sensing affected stem-like and exhausted T cells proportion *ex vivo*.”

However, it should be noted that *in vitro* priming of stem-like T cells to assess their proliferative capacity and stemness is standard in the field (PMID: 31827286, PMID: 30154266 for example), in part because of the rarity of the cells.

The anti-IFN γ should be performed in vivo to reach a convincing conclusion.

Because IFN γ is highly pleiotropic and can affect every cell in the tumor microenvironment, we decided it would be confounding to block IFN γ *in vivo*, as it could notably affect antigen presentation and tumor growth. As an alternative experiment, we initially transferred ad-mixed WT and IFN γ RKO OTI naïve cells into tumor-bearing mice, and analyzed their ratio in tumors after 5 days. We found an over-representation of IFN γ RKO cells (Fig.8D). In addition, we are now adding new data demonstrating that the enhanced expansion capacity in this system relies on stem-like T cells. To do so, WT and IFN γ RKO stem-like T cells were generated *in vitro*, ad-mixed, transferred in the same tumor-bearing mouse and analyzed after 6 days. We found that IFN γ RKO stem-like OTI cells had a greater expansion capacity compared to WT cells (Fig.5N), and had a greater propensity to retain the stemness (Tcf1 expression – Fig.6F-G). In addition, IFN γ RKO stem-like OTI cells are more potent in differentiating in cells with intermediate expression of exhaustion markers in tumors (Fig.S7H), which overcomes their primary defect in tumor homing (demonstrated in Fig.S2F). In the same conditions, the transfer of exhausted T cells revealed that IFN γ R deletion did not compensate for homing defects (Fig.5N).

Finally, we now also provide evidence that in human CD8 T cells undergoing checkpoint blockade, the expression of IFN γ R anti-correlates with multiple pathways related to cell division (Fig.1E-F).

3. In Fig 7, the authors concluded that the changes in exhausted cells are secondary to impaired stem-cell like activity. There is little evidence for this. The authors need to have lineage tracing studies to provide definitive proof. We agree that lineage tracing is the best way to confirm this hypothesis. However, there is no, to our knowledge, available reporter mouse to specifically and permanently label stem-like T cells, as all naïve cells express Tcf7. We would therefore need to inject a large number of tumor-infiltrating stem-like T cells in another tumor-bearing mice and follow stem-like T cells and their progeny over time. We attempted that experiment, but the rarity of those cells made it impossible in our hands. We estimated we would need to inject 0.5 million stem-like T cells per recipient mouse to confidently detect them after a few days, but we were able to isolate a maximum of 1,000 stem-like T cells from 4 tumors/mice. This experiment required far too many mice for this to be doable, at least in our hands. As an alternative tracing experiment, we generated WT and IFN γ RKO stem-like T cells *in vitro*, ad-mixed them, transferred them in tumor-bearing mice and analyzed their behavior in tumors after 6 days (see point 2 and Fig5L-N). Stem-like IFN γ RKO cells started to overcome the defect in migration and become dominant (Fig.5N), while retaining a greater proportion of stem-like T cells (Fig.6F-G) and generating a greater proportion of exhausted T cells with an intermediate expression of exhaustion molecules (Fig.S7I). Because IFN γ RKO *in vitro* generated exhausted T cells could not overcome the defect in migration (Fig.5N), we concluded that enhanced T cell expansion in CD8-IFN γ RKO mice likely results from unleashing stem-like T cells differentiation and downstream proliferation while enhancing their maintenance in tumors.

Our TCR sequencing data, although it lacks dynamic information, can also be used as lineage tracing. We observed a large overlap between stem-like T cells and exhausted T cell clones, and the first 20 clones of stem-like T cells represent over 75% of exhausted T cells (Fig.4F). This proves that exhausted T cells largely originate from stem-like T cells in our model.

To precisely test whether proliferation is affected by IFN γ at the stem-like T cell stage or later, we also isolated tumor-infiltrating exhausted and stem-like T cells using the TCF7-GFP mouse and stimulated them *ex-vivo* with anti-

CD3 and anti-CD28 to further analyze their capacity to proliferate. Stem-like T cells have an enhanced proliferative capacity upon restimulation compared to exhausted T cells (Fig.5G-I). The limited proliferation of exhausted T cells could not be rescued by blocking IFN γ (Fig.5J-K).

Overall, our cell-tracing, TCR-tracing, and ex-vivo experiments support the notion that changes of exhausted T cells are secondary to stem-like T cells following inhibition of IFN γ sensing.

4. While the scRNA analysis is sound, this is the sole set of data for mechanistic studies. Many changes are suggested based on transcripts, the authors made no attempt to validate any aspects of these, such as coinhibitory receptor expression, glycolysis activity.

We validated some of the scRNAseq data, such as subset proportion (Fig4C), proliferation pattern (Fig.5) and stem-like T cell retention following priming (Fig.6).

We have now validated the inhibitory receptor expression data (scRNAseq – Fig7G-I) by flow cytometry. We found that inhibition of IFN γ sensing led to the accumulation of a population of exhausted T cells with intermediate expression of exhaustion receptors, at the endogenous level (Fig.7J-M) and using OTI T cells (Fig.S7I).

Measuring glycolysis activity (and OXPHOS) with Seahorse has been challenging because of the small number of cells we were getting out of the tumor. We attempted to use methods such as SCENITH instead, but it gave inconclusive data. Using flow cytometry, we now provide evidence that CD8- IFN γ RKO cells do not have an intrinsic defect in metabolism. Indeed, mitochondrial function was not enhanced (Fig.7E and S7C), the BEC index (Bioenergetic cellular index) was also similar (Fig.7F) and the glycolysis rate was not significantly different between control and CD8-IFN γ RKO mice (as measured by Glut1 expression, Fig.S7D). The only notable difference we could detect was a decrease in LDHA expression, suggesting that CD8-IFN γ RKO cells relied less on anaerobic glycolysis (Fig.S7E). Altogether, we concluded that IFN γ RKO CD8 T cells do not have an intrinsic defect in metabolism, but rather reflect enhanced activity and environmental adaptation, as originally hypothesized.

5. The data of TSW119 is over-interpreted. The use of one inhibitor is not convincing enough to make a major conclusion about IFN γ signaling inhibits Wnt. In addition, TSW119 did not have a clear impact in promoting stem-like cells in the mock group (fig. 6i).

We agree that this was not enough, and we thank the reviewer for asking us to look into this, as it led to surprising data and, as such, we revised our conclusions. To address the cross-talk between IFN γ and Wnt, we used different inhibitors and agonist of the Wnt pathway and directly measure Wnt activity by quantifying b-catenin expression. Surprisingly, we could not detect any increase in b-catenin expression following Wnt agonist or TSW119, despite trying multiple doses, multiple time points as well as multiple T cell states (see for example Fig.S6D). This has already been documented (PMID: 20448567), and we honestly cannot find an explanation for this. However, we detected increased b-catenin expression following TCR priming, and this increase was restricted by IFN γ treatment (Fig.S6D-E). We therefore concluded that: “IFN γ might inhibit the intrinsic capacity of T cells to maintain a self-renewing pool following activation. Consistent with this, TCR-induced generation of stem-like T cells is further potentiated by anti-IFN γ treatment (Fig.S6F).” In this context, we hypothesize that Wnt generates stem-like T cells through a distinct mechanism independent of IFN γ .

6. Stem-like cells are usually more quiescent, compared with the exhausted counterpart. This quiescent state is necessary for its long-term maintenance. The KO stem-like cells showed more cycling, which may not be compatible with the interpretation that stemness was promoted in KO.

We agree that there is an apparent discrepancy, which has been observed previously (PMID: 31827286, PMID: 30154266). We believe this is in part because Stem-like T cells down-regulate TCF1/7 and up-regulate exhaustion markers rapidly after priming (PMID: 31827286). Some exhaustion markers are up-regulated as soon as the first division. As a result, most of the stem-like T cells are Ki67 negative (i.e they appear more quiescent), but they have enhanced proliferation capacity compared to exhausted T cells (PMID: 31827286, PMID: 30154266, PMID: 31810882, PMID: 30778252). As such, stemness and high proliferative capacity following restimulation are both the hallmark of stem-like T cells. Our data suggest that they are both limited by IFN γ (see points 2 and 3). We are now discussing this point in the manuscript:

“It is often assumed that IFN γ directly inhibits cytotoxic T cells (CTL) cytokinesis or kill CTLs. But CTLs and here, exhausted T cells, are barely responsive to IFN γ as they down-regulate IFN γ R2 (Fig.1A-B, 4D), critical for signaling. This has also been shown in the context of infection and in human CD8 T cells^{36, 74}. We discovered that IFN γ targets exhausted precursors, stem-like T cells, which quickly differentiate into an intermediate exhausted state upon restimulation, explaining how IFN γ indirectly affects exhausted T cells. Stemness and high proliferative capacity following restimulation, although counter-intuitive, are both hallmarks of stem-like T cells^{46, 73}. Our data suggest that they are both limited by IFN γ . This is consistent with the role of IFN γ in regulating the stemness of other cell types. IFN γ has a complex impact on hematopoiesis during inflammation⁷⁴, where, for instance, it impairs the maintenance of hematopoietic stem cells by directly inhibiting their proliferation and restoration upon viral infection⁷⁵.”

7. In fact, the alternative interpretation is IFN γ R1 KO cells as a whole have increased expansion capacity (with IFN γ as a limiting factor), in both exhausted and stem-like subsets (as evident in Fig. 2e, assuming the cell numbers are per gram of tumor, since the tumors are smaller in KO). Both subsets likely have increased glycolytic and oxphos capacity, decreased expression of coinhibitory receptors. Although KO cells did not increase cytotoxic cytokines, the numeric advantage and other aspect should have contributed to increased antitumor effect. In this process, the differentiation from stem-like to exhausted cells could be enhanced in KO, but direct evidence for this process is needed for this conclusion. Additionally, the KO cells infiltrated tumor better (Fig. 2g). Migration and tumor residency should be other mechanistic aspects that can be explored. The conclusion of exclusive impact on stem cells sounds more appealing, but the data suggesting a global impact on all cells on many functional readouts. The sole focus on stem-like cells does not seem to be justified.

We do agree that the phenotype induced by IFN γ R deletion in CD8 T cells is complex and multi-layered, and that increased cell number, decreased coinhibitory receptor and intra-tumoral localization in our CD8-IFN γ RKO mice all contribute to increased tumor control. There is a discrepancy between the expression of the IFN γ R, where IFN γ R2 is virtually absent in effector/exhausted T cells, and their apparent enhanced response (expansion) following IFN γ R ablation. We believe this discrepancy can be, at least in part, explained by the fact that IFN γ indirectly affects exhausted T cells by targeting their precursors. The reason we believe this is the case is: i) stem-like T cells have higher IFN γ R (especially R2) expression compared to exhausted T cells (Fig.4D), ii) lineage-tracing experiments (Fig.5L-N, Fig.6F-G, Fig.7H-I) and TCR sequencing (Fig.4F) demonstrate that stem-like T cells give rise to exhausted T cells, and if they cannot sense IFN γ , there is an increasing proportion of daughter cells with an intermediate exhaustion phenotype (Fig.S7I), iii) *ex vivo* restimulation demonstrate that blocking IFN γ has no direct effect on exhausted T cell proliferation (Fig.5J-K). Other studies have also found that activated T cells do not respond to IFN γ stimulation (PMID: 15905520). We hope the reviewer will agree that our data is now convincing, and that stem-like T cells are key to the phenotype of our CD8- IFN γ RKO mice. Importantly, as mentioned above, following the suggestion of the reviewer, we also tested whether tumor homing was enhanced by IFN γ R deletion. We found the opposite, where IFN γ R deletion impaired homing to tumors (Fig.S2F), most likely because of T cell failure to upregulate CXCR3 (Fig.S5H). We observed similar enrichment of WT over IFN γ RKO cells in tumors when we transferred exhausted T cells, however, enhanced proliferation induced by IFN γ R deletion compensated for this homing defect when stem-like T cells were transferred. This again, strongly suggests that IFN γ does not directly target exhausted T cells, at least for proliferation.

Finally, we would like to point out that increased proliferation and migration is not enough to explain increased clonal diversity observed in IFN γ RKO mice, which is, to our knowledge, also novel and highlights the fact that other mechanisms are at play.

minor:

1. the authors need to document that there is no aberrant activation of KO T cells, before tumor implant, so as to exclude secondary effect.

We activated WT and IFN γ RKO OTI with different altered peptides and could show there was no difference in priming, as exemplified by CD69 upregulation (Fig.8E). In addition, we now provide scRNAseq from tumor-draining LN of WT and CD8- IFN γ RKO mice, which shows that there is no difference in gene expression in CD8 T cells, supporting the notion that T cells do not have increased activation (Fig.S3C).

2. Statistics is missing, especially figure 1. Individual data point should be shown in all bar graphs. We apologize for this oversight, this has now been edited throughout.

Reviewer #2 (Remarks to the Author):

Mazet et al report on the functional impact of IFN γ signaling on tumor infiltrating CD8 T-cells. The premise of this study is based on earlier publications relating IFN γ signaling to expression of immune checkpoint molecules and CD8 T-cell exhaustion. IFN γ has been also correlated with increase tumor infiltration by T-cells and better clinical outcomes. The authors argue convincingly that the level of IFN γ signaling may be important in differentiating better or worse immune surveillance and disease outcomes. However, the study is focused on the genetic knock out of IFN γ R in CD8 T-cells. The authors use publicly available scRNAseq data from cancer patients and use mice transplanted with OVA expressing B16 melanoma cells. Two different strategies are used, tumor recipient mice have IFN γ R deficient CD8 T-cells or IFN γ R deficient OVA specific T-cells are adoptively transferred to tumor transplanted WT mice. Tumor growth and survival curves are suggestive of better protection by IFN γ R KO CD8 T-cells. Few mice and large variations within each cohort of mice are weak points.

We would like to thank the reviewer for his/her in-depth assessment of our manuscript, and pushing us to confirm our hypothesis. We hope the reviewer will agree that the manuscript and the message have greatly improved.

Presentation of data is not optimal, as results are often presented as bar diagrams not showing individual animals. We apologize for this oversight, this has now been edited throughout.

The authors perform scRNAseq analysis to better understand the molecular mechanism. While they claim an increase in the Stem like cluster of CD8 T-cells, most of the changes in gene expression that are discussed are in the exhausted population.

We do agree that there is an apparent discrepancy. We believe that this is in part due to the fact that, as soon as Stem-like T cells get primed, they down-regulate TCF1 and up-regulate exhaustion markers (PMID: 31827286 for example). As such, we believe that recently primed stem-like T cells rapidly transition to an intermediate exhausted pool, hence the variation we find in the exhausted cluster. To confirm this, we specifically sorted stem-like and exhausted T cells from WT tumour-bearing mice, restimulated those cells *ex vivo* in the presence of IFN γ blocking antibodies. Although exhausted T cells show the highest gene expression changes in cell cycle (Fig.S5G), they have clear decreased proliferation capacities compared to stem-like T cells experimentally upon restimulation (Fig.5G-I). Our data are in agreement with a large body of literature that reported the same enhanced proliferation of stem-like T cells (PMID: 31827286, PMID: 30154266, PMID: 31810882, PMID: 30778252). Importantly, restimulation of stem-like T cells induces the down-regulation of Tcf1/7 (Fig.6H) which was already reported and concomitant with increased expression of exhaustion markers (PMID: 31827286). We found the exact same result when we performed lineage tracing experiment using *in vitro* generated stem-like WT and IFN γ RKO OTI cells that we transferred in tumour-bearing mice and followed their progeny *in vivo*. After 6 days, 70% of WT OTI transferred lost Tcf1/7 expression (Fig.6F-G) and up-regulated exhaustion markers (Fig.S7H). Importantly, deleting or blocking IFN γ sensing in those experiments increased the maintenance of stem-like T cells (Fig.6F-H).

Overall, to highlight this, we concluded that: "Because IFN γ did not directly target exhausted T cells but most cells exhibiting transcriptional cell cycle signature belonged to the Exhausted/Bystander states (Fig.S5G), we concluded that cycling cells corresponded to newly differentiated progeny downstream of stem-like T cells."

The authors conclude that IFN γ R signaling leads to T-cell exhaustion, depletion from tumor, and activation induced cell death. Conclusions drawn are stretched and sometimes confusing. Validation experiments and some controls are missing. More comments on data presented in individual figures are below.

Apologies if the manuscript was not clear enough, but we did not want to conclude any of those.

i) We show that deletion of IFN γ R in CD8 T cell does not increase cell death (Fig6 C-D) both *in vitro* and *ex-vivo*.

ii) We do not conclude that IFN γ deplete all T cells at the tumor site. It inhibits the maintenance of stem-like T cells and cytokinesis of their progeny.

iii) scRNAseq shows that the average expression of exhaustion markers is lower in the IFN γ RKO compared to the WT CD8 T cells (Fig.7G-I). However, we do not claim that IFN γ signaling leads to exhaustion, but rather that this reflects the fact that IFN γ signaling inhibits stem-like T cells in tumors, reducing the flow of cells being differentiated on site: "Analysis of T cell clonality suggested that the increased self-renewal capacity of stem-like T cells in CD8-IFN γ RKO mice would perpetuate the generation of newly exhausted T cells, leading to an overall decreased exhaustion state without affecting exhaustion per se."

We edited the manuscript to remove unclear sentences. We also formally tested the latter point by performing lineage-tracing. To do so, WT and IFN γ RKO stem-like OTI cells were generated *in vitro*, ad-mixed, transferred in the same tumor-bearing mouse and analyzed after 6 days. We found that IFN γ RKO stem-like OTI cells had a greater expansion capacity compared to WT cells in the tumor (Fig.5N), and a greater propensity to retain stemness phenotype (Tcf1 expression – Fig.6F-G). In addition, IFN γ RKO stem-like OTI cells are more potent in differentiating in cells with intermediate expression of exhaustion markers in tumors (Fig.S7I), which overcomes their primary defect in tumor homing (demonstrated in Fig.S2F). In the same conditions, the transfer of exhausted T cells revealed that IFN γ R deletion did not compensate for homing defects (Fig.5N).

In the first Figure the authors use publicly available scRNAseq data from an earlier report by Watson et al 2021, to show lower expression of IFN γ R1 and IFN γ R2 in cycling cells tumor infiltrating T-cells. The Watson et al study involved the analysis of 4 responding and 4 non-responding metastatic melanoma cancer patients who had received immune checkpoint blockade treatment; in each set 2 had received only Pembro and 2 Nivo + Ipi. The figure is lacking critical information and is thin in analysis. The bar diagrams do not include statistical evaluation and do not distinguish the subgroups of patient.

We apologise for this oversight and have now added all relevant statistics to the plots.

Apologies again if the manuscript was not clear enough. In the study from Watson et al, scRNAseq metadata does not have any information on the patient response to checkpoint blockade. Eight patients who were prescribed ICB for Metastatic Melanoma at the Churchill Hospital, Oxford University Hospitals NHS Foundation Trust, UK were prospectively and sequentially recruited. Four participants received single agent pembrolizumab (sICB) and four received combination ipilimumab/nivolumab (cICB). Even if we had follow-up clinical outcomes, we believe the data would have been underpowered to perform the analysis suggested.

In this figure the authors also have box diagrams that show lower expression of the receptors in larger T-cell clones, based on the clonal analysis by Watson et al using TCR sequencing. The Watson report however indicates that the larger clones are non-cycling. This produces therefore a disconnect between lower expression of the receptors and the cycling status of the T-cells, which weakens the authors' conclusion that lower expression of the receptors is confined to the cycling stem like T-cells.

Watson et al reported that: "large EC clones had a notable divergent gene expression profile after ICB compared with both small EC clones (Watson, Science Immunology 2021, Fig. 3F) and large EM clones (Watson, Science Immunology 2021, Fig. 3G), with pathways including T cell activation, **proliferation** and costimulation, TCR signaling, and IFN γ production uniquely up-regulated in large ECs." EC being "Effector Clones" and EM corresponding to "Effector Memory". While the large clones preferentially fall into the Effector cluster rather than the small highly-mitotic cluster in the blood, it does not mean they do not proliferate at the tumor site or in LNs. GO analysis suggests they indeed have a proliferation signature and DEG expression demonstrates enhanced proliferation compared to smaller clones (Watson, Science Immunology 2021, Fig.3F). This is therefore not in contradiction with our mouse data showing enhanced proliferation when IFN γ R is deleted. This is also supported by new analysis performed on the bulk RNAseq data (see below).

Deeper analysis of the data would have helped. For example, does expression of IFN γ receptors relate to downstream signaling across different CD8 T-cell subtypes, patients, and response to treatment? Does expression of IFNR and IFN γ signaling relate to genes that Watson et al found to be consistently down or upregulated with therapy response such as IL10R and GZMA.

Watson et al found that: “Genes such as IL10RA and GZMA were consistently down- and up-regulated by treatment (in the Effector cluster specifically)”, not by response to therapy as suggested by the reviewer. Although we could not address response to treatment in this dataset (see above), we investigated the general correlation between IFN γ R, signaling and IL10R/GZMA expression as suggested. We used the ‘correlatePairs’ function from the simpleSingleCell package. This has a statistical method to generate a p-value for significance of the Spearman rho between two genes, taking into account noise. While the correlation mostly fits the hypothesis (decreased IFN γ R2 and increased GzmA following blockade, for example, leading to anti-correlation between both genes), it was becoming clear that looking at a few specific genes was adding only limited information.

```
var.cor %>% filter(gene1 %in% c('IFNGR1','IFNGR2'), FDR<0.05)
```

gene1	gene2	rho	p.value	FDR	limited
IFNGR2	GZMA	-0.06238997	1.999998e-06	1.559998e-05	TRUE
IFNGR1	GZMA	0.03537429	1.999998e-06	1.559998e-05	TRUE
IFNGR1	TYROBP	0.02483240	2.079998e-04	1.247999e-03	FALSE

However, we agree with the reviewer that we have not taken full advantage of this data. We, therefore, included another analysis in the manuscript which characterized the potential consequences of IFN γ R dynamic expression by analyzing the genes correlated with the expression of IFN γ R1 and R2 (Table S1 and S2) from bulk RNAseq data after treatment. Using normalized IFNGR1/2 expression as a continuous variable in DESeq2, we revealed that both chains are anti-correlated with pathways linked to cytokinesis and migration (Fig.1E-F). In addition, consistent with the fact that IFN γ R1 and R2 expression are differentially regulated on different subsets, they are also associated with some distinct pathways. For example, IFN γ R2 anti-correlated with pathways related to cytotoxicity (Fig.1F), in agreement with the fact that IFN γ R2 was downregulated after checkpoint blockade and on large clones, as both conditions are characterized by increased cytotoxicity.

Overall, this new analysis strengthens the relationship between IFN γ R expression and clonal size, but also later data obtained experimentally on proliferation (Fig.5), migration (Fig.S2F), and stemness (Fig.1F and Fig.6).

Also, it would be helpful to provide information on the patient cohort, treatments, and outcomes rather than just referring to the publication of Watson et al.

This is indeed important, and most likely the source of some confusion. We apologize for this and have now included more information on the cohorts to avoid confusion as such:

“Single-cell RNAseq data from Watson et al was analyzed for IFNGR1/2 expression across labelled CD8 T cell subsets or clone size groupings. The cohort consisted of eight patients who were prescribed checkpoint blockade. Four participants received single agent pembrolizumab and four received combination ipilimumab/nivolumab. Blood samples were collected immediately before treatment at d0 and d21 after 1 cycle of treatment. CD8 T cell subset annotations were kept consistent with the annotated Seurat object from Watson et al, with the exception of MAIT cells, which were further annotated as cells carrying TRAV1-2 TRAJ12/20/33 TCR chains within MAIT-containing Seurat clusters (1, 19, 20 and 24). Clone size groups were calculated as a proportion of the TCR repertoire per sample, in accordance with the original definitions. Cells with above-zero normalized expression of IFNGR1/2 were defined as IFNGR1/2-expressing.

Expression of IFNGR1/2 in bulk RNAseq data from CD8 T cells was analyzed using data from Watson et al (EGAS00001004081). Samples consisted of blood CD8 T cells from individuals taken at 21 days after treatment (n=110) for which clinical follow-up was available, as described in Watson et al. Expression of IFNGR1/2 was analyzed using DESeq2-normalized expression data. To identify pathways that correlated with IFNGR1/2 expression, normalized IFNGR1/2 expression was used as a continuous variable in DESeq2. Comparisons between individuals with divergent 6-month progression and autoimmune toxicity outcomes were performed using Wilcoxon rank sum tests.”

In the second figure the authors compare the growth of transplanted tumors in mice with IFNR deficient or sufficient CD8 T-cells and relate to densities of tumor infiltrating CD8 T-cells. Based on the data presented, the authors conclude that IFNgR deficiency increases tumor infiltration by CD8 T-cells and dampens the kinetics of tumor growth. There are several concerns with the presented results. Results from a single experiment with up to 7 mice per cohort

is presented although the authors state that the data is representative of 3 independent experiments. It is unclear why all three experiments are not shown and statistically analyzed.

Survival curves are shown as percent survival which does not allow evaluation of the fate of individual mice. With only 7 mice per group the difference in survival of the two groups is less than convincing.

In two separate plots the authors show average tumor volume and the tumor volume of individual mice over time. It is clear from the latter plot that only 3/7 control mice show more rapid tumor growth than the IFN γ R deficient mice. This makes the 3-stars significance for the difference in average tumor growth in the two cohorts of mice rather surprising. Tumor weight difference is shown as bar diagram, this time cumulative results of 3 different experiments; individual tumor weights are not shown. The absolute number of tumor-infiltrating CD8 cells per tumor are plotted as bar diagram, but individual values are not shown. CD8 T-cells are also presented in relation to tumor volume. Here the number of mice analyzed is less than 7 per group.

We now have cleaned up and consolidated our data.

We have now pooled the 3 experiments as suggested and added more mouse data where possible (n=12 for Control and n=13 for CD8-IFN γ RKO for Fig.2A-C; n=18 for Control and n=16 for CD8-IFN γ RKO for Fig.2D).

The survival result is statistically significant based on Gehan-Breslow-Wilcoxon test.

For the tumour volume over time, multiple comparisons were performed with a Šidák's test in Graphpad prism.

We are happy to provide the files if required.

We now present data as scatter plot instead of bar plot throughout.

Spatial distribution of CD8 cells is shown in IF stained section, and percent of CD8 cells in the tumor is plotted as bar diagrams not showing counts from individual tumors.

We now show all our data as scatter plot, each point here corresponds to a tumor.

The authors measure secretion of proinflammatory cytokine after stimulating the tumor infiltrating CD8 T-cells with OVA peptide and find no difference. They conclude that, "IFN γ R ablation in CD8 T cells does not substantially enhance intrinsic functional properties but it increases CD8 T cell infiltration to the core of the tumor. that IFN γ sensing by CD8 T cells is an independent pathway restricting T cell anti-tumor immunity". This conclusion is stretched and hard to justify. The most common mechanism of tumor rejection by CD8 T-cells is cytotoxicity not cytokine secretion.

Inflammatory cytokines have been implicated in tumour control in many studies. For example, IFN- γ can exert direct cytotoxic or cytostatic effects on tumour cells (PMID: 8614832, PMID: 11900986), and contribute to tumour senescence (PMID: 23376950) and tumour ferroptosis (PMID: 31043744). We previously investigated cytotoxicity in the OTI system (Fig.8H-I), where we could not detect any difference. To further our analysis and directly answer the reviewer's comment, we also addressed this in our CD8- IFN γ RKO mice by analysing surface LAMP1 on CD8 T cells, which is a readout of T cell degranulation (Fig.2I). For consistency, we also performed that experiment on OTI cells (Fig.8J). Both sets of experiments revealed that IFN γ R deletion does not impair cytotoxicity.

Direct evidence for increased migration of IFN γ R-KO T-cells to the core of the tumor is lacking.

We apologise for the lack of clarity and explanation on this issue. Differential location between control and IFN γ RKO TILs can be the result of many mechanisms, including increased chemotaxis, recruitment but it could also reflect interactions with other cells, the state of the T cell, or simply reflect active killing. We edited the text to convey the fact that differential localization is not necessarily due to increased migration. We also investigated the role of IFN γ signaling for T cell recruitment. WT and IFN γ RKO OTI cells were activated *in vitro*, ad-mixed (1:1 ratio), and transferred in tumor (B16-OVA) bearing mice. The ratio between WT and IFN γ RKO OTI was analyzed by flow cytometry after 24h. We found a strong defect in migration in IFN γ RKO OTI cells (Fig.S2F). We had similar data when we transferred WT and IFN γ RKO exhausted OTI T cells and analysed ratio in tumours after 6 days (Fig.5L-N).

Another sentence which seems at odds with published literature is, "Recent data suggests that the deeper infiltrating TILs are further along the exhaustion/dysfunctional pathway compared to the T cells at the margin". Indeed, studies by Schietinger and others demonstrate that exhaustion happens early during tumor onset and it is the reversibility of the exhaustion phenotype that is affected over time.

The statement we made was coming from 2 recent papers (Kersten, Cancer Cell 2022, Hu, Nat. Methods. 2020). We were not arguing there was any temporal aspect to this, just that localisation can be influenced by the state of the T cell. We therefore removed this sentence to avoid any confusion.

Results from cytokine assays are shown as bar diagrams, without individual values. Controls are missing, e.g unstimulated, stimulated with an irrelevant peptide, same tumor without OVA. B16 melanoma cells are known to downregulate MHC-I and to metastasize to the draining lymph nodes. Checking expression of MHC and involvement of the draining lymph nodes would help.

We have now included individual values and included unstimulated controls.

To our knowledge, subcutaneous injections of B16 do not metastasize in the time frame we are using, which is most likely too short to allow for this to occur. Consistent with this, MHC-I is up-regulated on B16-OVA, (but not B16-OVA- IFN γ RKO, used here as a control), 12 to 15 days post-enuftment in WT mice.

Finally, scRNAseq of CD8 T cells from draining LN of ctrl and CD8- IFN γ RKO shows that there are no differentially regulated genes between WT and IFN γ RKO CD8 T cells (Fig.S3), also suggesting that most of the effect of IFN γ is confined to tumours in those conditions.

Figures 3 and 4 show results of scRNAseq analysis of tumor infiltrating Ova specific CD8 T-cells. Up to eight clusters expressing different levels of exhaustion markers are identified. Using differential analysis they show that the stem like cluster of WT CD8 T-cells has higher expression of IFN γ R than the other clusters, and that a higher proportion of WT T-cells express exhaustion markers as compared to IFN γ R deficient T-cells. Using TCR clonal analysis, the stem like T-cells share TCRs with the other subsets but have smaller clonal populations. Using pseudo-timing they provide results suggesting that expansion of control T-cells occurs early while expansion of IFN γ R deficient T-cells occurs throughout the trajectory to exhausted clusters. The authors conclude that "TILs from CD8-IFN γ RKO mice contained an increased proportion of cells belonging to the stem-like cluster". The conclusion needs to be validated at the protein level using high dimensional flow cytometry.

As a side note, the scRNAseq was performed on endogenous WT and IFN γ RKO CD8 T cells coming from B16OVA tumors and their draining LNs (not on OVA-specific CD8 T cells). The greater stem-like T cells proportion in IFN γ RKO CD8 mice found in the scRNAseq data was validated by flow cytometry, figure 4 panel C. In addition, we have now strengthened this aspect, as it is key to our message. We now provide more direct evidence that IFN γ inhibits stem-like T cell maintenance (Fig.6). First, we analyzed stem-like signatures in our mouse scRNAseq data and found that CD8-IFN γ RKO mice displayed enhanced stem-like signature (Fig.6E). We also performed lineage-tracing experiments. *In vitro* generated WT and IFN γ RKO OTI stem-like T cells were transferred in tumor-bearing mice. After 6 days, the percentage of OTI cells that retained a stem-like T cell phenotype (PD-1⁺ TCF7⁺) was analyzed by flow cytometry. IFN γ RKO OTI consistently had a higher percentage of cells that retained a stem-like phenotype (Fig.6F-G). Similarly, we isolated intra-tumoral WT stem-like T cells and restimulated them *ex vivo*. Flow cytometry

analysis revealed that when IFN γ is blocked, CD8 T cells better retained TCF1/7 expression compared to controls (Fig.6H).

Figures 5 and 6 analyze cell cycle and stem cell signatures of the OT1 T-cells from scRNA, comparing IFN γ R sufficient and deficient OT1 cells after adoptive transfer to B16-OVA tumor transplanted mice. It is not clear where the cells were harvested from, blood, tumor DLN, or tumor.

Apologies for this oversight, OTI cells (from new Fig.S5C-D) were harvested from the tumour. We edited the manuscript to make this clear. As a side note, scRNAseq has been done on endogenous CD8 T cells, not OTI.

Surprisingly, cells in the exhausted clusters show the strongest cell cycle signature.

We agree that there is an apparent discrepancy, which has been observed previously (PMID: 31827286, PMID: 30154266). We believe this is in part because Stem-like T cells down-regulate TCF1/7 and up-regulate exhaustion markers rapidly after priming (PMID: 31827286). Some exhaustion markers are up-regulated as soon as the first division. As a result, most of the stem-like T cells are KI67 negative (i.e they appear more quiescent), but they have enhanced proliferation capacity compared to exhausted T cells (PMID: 31827286, PMID: 30154266, PMID: 31810882, PMID: 30778252). As such, stemness and high proliferative capacity following restimulation are both hallmarks of stem-like T cells. Our data suggest that both stemness maintenance and increased proliferative capacities are limited by IFN γ . To precisely test whether proliferation is affected by IFN γ at the stem-like T cell stage or later, we also isolated exhausted and stem-like T cells from tumors using the TCF7-GFP mouse and stimulated them *ex-vivo* to analyse their capacity to proliferate. Stem-like T cells have an enhanced proliferative capacity upon restimulation compared to exhausted T cells (Fig.5G-I). Limited proliferation of exhausted T cells could not be rescued by blocking IFN γ (Fig.5J-K).

Finally, in another set of experiments, we confirmed the fact that stem-like T cell expansion is more sensitive to IFN γ than exhausted T cells. We generated exhausted CD8 T cells that were either WT or IFN γ RKO, that we admixed and transferred in a tumour-bearing mice. After 6 days, we analysed the ratio between WT and KO transferred cells and, consistent with the defect in migration of IFN γ RKO cells, we found 5X more WT than IFN γ RKO exhausted T cells in tumours (Fig.5L-N). However, when we performed the same experiment using *in vitro* generated stem-like T cells, IFN γ RKO cells started to overcome the defect in migration and half of the tumours displayed more IFN γ RKO than WT transferred cells (Fig.5N).

We are now discussing this point in the manuscript in two distinct sections:

Page 7: "Because IFN γ did not directly target exhausted T cells but most cells exhibiting transcriptional cell cycle signature belonged to the Exhausted/Bystander states (Fig.S5G), we concluded that cycling cells corresponded to newly differentiated progeny downstream of stem-like T cells."

Page 10: "It is often assumed that IFN γ directly inhibits cytotoxic T cells (CTL) cytokinesis or kill CTLs. But CTLs and here, exhausted T cells, are barely responsive to IFN γ as they down-regulate IFN γ R2 (Fig.1A-B, 4D), critical for IFN γ -downstream signaling. This has also been shown in the context of infection and in human CD8 T cells^{36, 74}. We discovered that IFN γ targets exhausted precursors, stem-like T cells, which quickly differentiate into intermediate exhausted state upon restimulation, explaining how IFN γ indirectly affects exhausted T cells. Stemness and high proliferative capacity following restimulation, although counter-intuitive, are both hallmarks of stem-like T cells^{46, 73}. Our data suggest that they are both limited by IFN γ . This is consistent with the role of IFN γ in regulating stemness of other cell types. IFN γ has a complex impact on hematopoiesis during inflammation⁷⁴, where, for instance, it impairs the maintenance of hematopoietic stem cells by directly inhibiting their proliferation and restoration upon viral infection⁷⁵."

The analysis of stem cell genes focuses on expression of heat shock proteins by the stem like cluster rather than the exhausted cluster, which gains the highest number of cycling cells with loss of IFN γ R. There is therefore a disconnect between these two figures and some confusion about the relevance of stress to stem cell signature and cell cycle signatures. The confusion is confounded by the concluding sentence: "These data indicate that stem-like T cells might get depleted at the tumor site because of IFN γ -induced stress and/or inhibition of cytokinesis". No data is provided to support this conclusion.

Figure 6 analyses stem cell markers and show elevated expression of heat shock genes and cell cycle genes . That this implies stem like characteristics is hardly convincing. More likely it reflects cell stress due to absence of INFgR signaling. Data is presented indicating that stimulation of T-cells in the presence of INFg does not increase cell death, which is at odds with some of the assumptions put forward in the manuscript.

We focused on heat shock molecules in the stem-like cluster because HSP have been implicated in stemness (see manuscript), and the stem-like cluster expresses higher transcript of HSP compared to the exhausted cluster (Figure S6A). HSP have been implicated in stemness in multiple cell types (see manuscript for details) and were simply used as markers here. To strengthen the relative stemness of WT vs CD8- IFN γ RKO cells, we are now comparing stem-like signature scores (Fig.6E) and found that IFN γ RKO cells had enhanced stem-like signature scoring. We also directly tested the function of IFN γ in stemness *ex vivo* and lineage tracing experiment, as depicted above (Fig.5 and Fig6). We hope the reviewer will now agree with our conclusion that IFN γ sensing by CD8 stem-like T cells restricts their maintenance.

In addition, we do not hypothesize that increase cell cycle genes is a stemness characteristic, although it has been shown in multiple studies that stem-like T cells have enhanced proliferative capacities following restimulation compared to exhausted T cells. This discrepancy (difference between cell cycle signature and proliferation capacity after restimulation) has been observed previously by multiple independent labs (PMID: 31827286, PMID: 30154266, PMID: 31810882, PMID: 30778252). For example, a publication from Rafi Ahmed lab (PMID: 27501248 shows that exhausted T cells have higher cell cycle gene signature compared to stem-like T cells (Extended Fig.5), although stem-like T cells have a higher expansion rate when transferred in new hosts (Fig.3a-c) and specifically provide the proliferative burst after PD-1 therapy (Fig.3f-g). We believe this is similar to what we observe in our model. We will review the text to confirm there is no confusion.

In line 270 the term “IFNgRKO ablation” needs to be corrected.

We edited this mistake.

More correlations are shown using publicly available data, indicating increased expression of TCF1 and lower expression of INFgR by T-cells of cancer patients. The assumption that expression of TCF1 alone denotes stem cell like T-cells is an overstatement.

We have now used the stem-cell signature from (Pace et al, 2018) instead, which shows similar correlation (Fig.6I). We have also applied a stem-like signature to our mouse dataset, which confirms that inhibition of IFN γ sensing in CD8 T cells leads to increased stem-like signature scoring (Fig.6E). In addition, bulk analysis of metastatic melanoma patients undergoing checkpoint blockade shows an inverse correlation between IFN γ R2 expression and pathways related to stemness maintenance (Fig.1F).

Figure 7 deals with the metabolic signature of the T-cells. The INFgR KO T-cells exhibit more glycolysis and more Oxphos metabolism than control T-cells (Fig 7 C-F), suggesting overall higher metabolism. The authors conclude with confusing statements: “...the increased self-renewal capacity of stem-like T cells in CD8-IFN γ RKO mice would perpetuate the generation of newly exhausted T cells, leading to an overall decreased exhaustion state without affecting the path of exhaustion per se” and, “these data are consistent with a steady generation of TILs in CD8-IFN γ RKO compared to control mice. This could also explain the subtle increase in cytokine production observed experimentally in IFN γ RKO CD8 T cells (Fig. 2) and suggested it could be driven by a subpopulation of TILs that recently got generated”. The scRNAseq data suggests an increase in the stem like cluster and not exhausted clusters, and there is no evidence that IFNgR KO T-cells at the core of the tumor are less exhausted that their WT counterparts”. Changes in metabolism of T-cells have not been mechanistically connected to increased tumor infiltration.

Figure 2 demonstrates that we have an overall increase of all T cells at the tumour site, which implies that exhausted T cells are indeed increased in the CD8-IFN γ RKO mice, since they represent 90% of all CD8 T cells in the tumour. The scRNAseq, however, suggests that within all those expanded cells, stem-like T cells occupy a larger proportion (in IFN γ RKO compared to WT). As such, we believe our data is not contradicting itself.

We did not infer any relationship between localization and metabolism. We have reviewed the manuscript to remove any confusing sentences.

In addition, we have confirmed experimentally the lesser level of exhaustion observed in our transcriptomics data. We found that inhibition of IFN γ sensing led to the accumulation of a population of exhausted T cells with intermediate expression of exhaustion receptors, at the endogenous level (Fig.7J-M) and using OTI T cells (Fig.S7H). Measuring glycolysis activity (and OXPHOS) has been challenging because of the small number of cells we were getting out of the tumor. We attempted to use methods such as SCENITH instead, but it gave inconclusive data. Using flow cytometry, we now provide evidence that CD8- IFN γ RKO cells do not have an intrinsic defect in metabolism. Indeed, mitochondrial function was not enhanced (Fig.7E and S7C), the BEC index (Bioenergetic cellular index) was also similar (Fig.7F) and the glycolysis rate was not significantly different between control and CD8-IFN γ RKO mice (as measured by Glut1 expression, Fig.S7D). The only notable difference we could detect was a decrease in LDHA expression, suggesting that CD8-IFN γ RKO cells relied less on anaerobic glycolysis (Fig.S7E). Altogether, we concluded that IFN γ RKO KO CD8 T cells do not have an intrinsic defect in metabolism, but rather reflect enhanced activity and adaptation in the microenvironment, as originally hypothesized.

Figure 8 uses adoptive T-cell transfer to document better protection by IFN γ R KO T-cells. This figure suffers from similar shortcomings as Figure 2, regarding the small number of animals examined and wide variations in survival of the animals tested.

We have now increased the number of animals in panels as such:

Fig.8A - n=7 control, n=17 WT and n=18 IFN γ RKO

Fig.8B - n=18 WT and n=18 IFN γ RKO

Fig.8C - n=9 WT and n=10 IFN γ RKO

It is interesting that IFN γ R KO mice produce more IFN γ than control T-cells, indicating negative feedback regulation which however is not further analyzed.

We agree that this is interesting. However, our *in vivo* data demonstrated that this had little effect on the overall IFN γ availability in the tumour (Fig.2H), suggesting that this is not the most important mechanism regulating anti-tumour immunity in our model and therefore beyond the scope of this paper.

Peer Review File

Reviewer comments further:

Reviewer #1 (Remarks to the Author):

The authors made good faith efforts to address this reviewer's concerns. The new experimental data and analyses further strengthen the authors' conclusion, and the revised interpretation is appropriate. Although the use of in vitro differentiated "exhausted" cells was not ideal, the reviewer recognizes the limiting factors, in particular cell numbers required for in vivo lineage tracing. Considering multiple lines of evidence that were provided, the authors' conclusions are reasonably well supported. The new knowledge warrants publication in Nature Communications.

Reviewer #2 (Remarks to the Author):

This revised manuscript present original and novel data detailing the mechanisms by which IFN-gamma signaling on CD8 T cells affects antitumor immunity. The authors carefully addressed all the reviewers concern by improving data presentation, clarifying unclear conclusions and most importantly by adding new data that directly address the question whether IFN affect stem like T cells in the tumors. The in vivo transfer/ lineage tracing new data add strength to the authors' conclusions and address reviewers' concern. Murine data now show the results with individual mice as rightly suggested by the reviewers. Reading through the revised manuscript, it appears that many of the inaccuracies in data presentation that have led the reviewers to misinterpret some of the authors' intended conclusions have been fixed. This together with the addition of many new data following reviewers' suggestions strengthen the manuscript. The novelty of the message is important, and the data are now well presented for the reader to understand and evaluate the authors' conclusions.